

# The impact of a new high-resolution ocean model on the Met Office North-West European Shelf forecasting system

Marina Tonani[1], Peter Sykes[1], Robert R. King[1], Niall McConnell[1], Anne-Christine Pequignet[1], Enda O'Dea[1], Jennifer A. Graham[2], Jeff Polton[3], and John Siddorn[1]

[1]Met Office, Exeter, EX1 3PB, UK
[2]Centre for Environment, Fisheries and Aquaculture Science, Lowestoft, NR33 0HT, UK
[3]National Oceanography Centre, Liverpool, L3 5DA, UK

*Correspondence to*: Marina Tonani (marina.tonani@metoffice.gov.uk)

**Abstract**

The North-West European shelf ocean forecasting system has been providing oceanographic products for the European continental shelf seas for more than fifteen years. In that time several different configurations have been implemented, updating the model and the data assimilation components.

The latest configuration to be put in operations, an eddy resolving model at 1.5 km (AMM15), replaces the 7km model
(AMM7) that has been used for a number of years. This has improved the ability to resolve the mesoscale variability in this area. An overview of this new system and its initial validation is provided in this paper, highlighting the differences with the previous version.

Validation of the model is based on the results of two years (2016-2017) trial experiments run with the low and high resolution systems in their operational configuration. The 1.5 km system has been validated against observations and the low resolution
system, trying to understand the impact of the high resolution on the quality of the products delivered to the users. Although the number of observations is a limiting factor, especially for the assessment of model variables like currents and salinity, the new system has been proven to be an improvement in resolving fine scale structures and variability and provides more accurate information on the major physical variables, like temperature, salinity and horizontal currents. AMM15 improvements are evident from the validation against high-resolution observations, available in some selected areas of the model domain.
However, validation at the basin scale and using daily means penalised the high-resolution system and does not reflect its superior performance. This increment in resolution also improves the capabilities to provide marine information closer to the coast even if the coastal processes are not fully resolved by the model.

## 1. Introduction

The North-West European Shelf (NWS) is a shallow shelf region consisting of the North Sea, the Irish Sea, the English Channel
and the surrounding waters of the Skagerrak, Kattegat in the east and the North and South-West approaches in the west. These shelf seas are predominantly shallow (with the exception of the Norwegian Trench) and highly tidal. Marine industries in these waters are substantial, with well-established fishing and commercial oil and gas industries more recently being joined by the renewables activities which are continuously expanding. The countries that have coastlines in the NWS are in the main densely populated in these coastal regions and so there are also significant populations that are directly affected by the marine
environment in the NWS, with coastal flooding a particular issue due to the high tides, waves and storm surges. The increasing focus on understanding the marine environment in support of sustaining healthy and biologically diverse seas is also a considerable driver in these waters, where heavy industrial and farming activity, as well as fishing and climate change, may have significant impacts on the quality of the marine environment.





There is therefore a significant history of marine monitoring and prediction in support of sustainable use of our marine environment, with the Safety of Life at Sea imperative leading to surface wave models providing forecasts, followed by ocean model forecasts predominantly by the need for storm surge prediction and (more recently) currents and hydrodynamics, in the main led by Defence requirements, but also supporting industry and marine planning. Most recently of all there has been an

increasing focus on sustained monitoring and forecasting of the lower trophic ecosystem and marine biogeochemistry. The Met Office, with the support of collaborators from around the North-West Shelf region, has for a number of years being producing freely available marine predictions and forecasts for this region as part of the Copernicus Marine Environment Monitoring Service (CMEMS, Le Traon et al, 2017) and precursor projects (e.g. Siddorn et al., 2007; O'Dea et al., 2012).

The operational ocean forecasting systems developed with the Forecasting Ocean Assimilation Models (FOAM) are based on a seamless prediction philosophy whereby the global and regional systems rely on similar ocean modelling and assimilation tools, and are co-developed for short-range forecasting, seasonal forecasting (e.g. MacLachlan et al., 2015, Tinker et al. 2018) and climate predictions (Williams et al, 2018). The operational forecasting configuration for the North-West European Shelf (NWS) is a FOAM system designed to deal with the specific constraints of operational oceanography on a shallow continental

shelf sea. The model domain (shown in Figure 1) extends into the Atlantic Ocean to resolve exchanges across the shelf, of primary importance for the continental shelf seas dynamics and water properties. The Atlantic region is chosen to allow the propagation and downscaling onto the shelf of phenomena associated with the large scale open ocean circulation. For example, the North Atlantic Current and European Slope Current which transport heat and salt from the North-East Atlantic, interact with the continental shelf slope and forms branches that flow into the North Sea. The boundaries in the Baltic cover the

Kattegat-Skagerrak area, to provide the Baltic inflow, which has a significant influence on the region's water masses due to the significant, and highly variable, freshwater fluxes.

The NWS system has three major components: an ocean model coupled with a biogeochemistry model and a variational data assimilation scheme. This system runs a forecast cycle every day to provide 6-day forecasts of the physical and biogeochemical

variables in this area. The forecast is initialized by running two 24-hr analysis cycles. By assimilating observations in this way, the FOAM system incorporates information from considerably more observations than would be available in near-real time with a single 24-hour window, due to the addition of late-arriving observations.

Until recently the operational system for this region has been run at approximately 7 km horizontal resolution (O'Dea et al.,

2012; King et al., 2018). This paper describes the operational implementation of the 1.5 km version of this ocean model, referred to as the Atlantic Margin Model, or AMM15 (Graham et al, 2018a). The dominant dynamical scales decrease with reducing water depth and have complex interactions with tidal phenomena and other bathymetric interactions requiring a modelling system at a kilometric scale resolution for this region (Polton, 2015, Holt et al., 2017). The increase of resolution to 1.5 km is therefore a fundamental step change in the ability to resolve key processes and features in the NWS region

(Guihou et al., 2017). As well as being developed for ocean forecasting operations, the AMM15 is being used within a coupled ocean-wave-atmosphere research system (Lewis et al., 2018a and 2018b).

The upgrade of the NWS system to AMM15 does not yet include the biogeochemical component as the computational cost is prohibitive. Two systems are therefore being run in parallel: i) the 7 km AMM7 model with the physical and biogeochemical components similar to O'Dea et al. (2012) and ii) a 1.5 km AMM15 physics only system (being described in this paper). An

AMM15 based coupled physics-biogeochemistry model is under development, and techniques are being developed to reduce the computational cost to allow it to be implemented within the time constraints of operational production. Herein, therefore we describe only the physical component of the high-resolution (AMM15) system, highlighting the differences with the low resolution configuration (AMM7). O'Dea et al. (2017, 2012) provide full descriptions of the AMM7 version of this system. It



is worth noting here that the NWS system also has a wave component providing products on the same grid as the AMM15. The wave and physical models are forced by the same wind fields and the wave model uses the surface currents computed by AMM15. However, the wave model description and validation are not within the scope of this paper. Here we provide details on the AMM15 hydrodynamic model, the data assimilation scheme, and the operational suite in the following section. Section

3 will then describe the trial experiments while Section 4 details our assessment of the new high-resolution products. Our conclusions are presented in Section 5.

## 2. System Description

### 2.1 Core Model Description

The Forecasting Ocean Assimilation Model (FOAM) 1.5 km Atlantic Margin model (AMM15) is a hydrodynamic model, one-

way nested within the Met Office operational North Atlantic 1/12° deep ocean model (Storkey et al. 2010) and the Copernicus operational Baltic Sea model (Berg et al. 2012.). The model core is based on version 3.6 of the Nucleus for European Modelling for the Ocean (NEMO, Madec 2016). This is a community ocean modelling system that has a wide user and developer base, particularly in Europe.

The regional model is located on the North-West European continental Shelf (NWS), extending from approximately 45°N to 63°N, and from 20°W to 11°E. There is a uniform grid spacing of ~1.5 km throughout, in both the zonal and meridional direction (Graham et al. 2018a). The vertical coordinate system is based on a hybrid s-sigma terrain following system, z *− σ (Siddorn and Furner, 2013), with 51 vertical levels. This is the same as that used in AMM7, with the thickness of the surface cell set to ≤ 1 m to guarantee uniform surface heat fluxes across the domain. The terrain-following coordinates used here are

fitted to a smooth envelope bathymetry, where the level of smoothing is determined such that the local bathymetric slope $r = (h_i − h_{i+1})(h_i + h_{i+1})$, computed between adjacent bathymetry points $h_i$ and $h_{i+1}$, is constrained to be less than a specified maximum value ($r_{max}$). This is required to mitigate against spurious horizontal velocities that arise from horizontal pressure gradient errors in terrain following coordinates that are too steep. Although the number of levels in both AMM15 and AMM7 are the same, the steeper gradients resolved in AMM15 means that a lower $r_{max}$ value was chosen (0.1, compared to 0.24) to

ensure stability along the shelf-break.

The bathymetry chosen for AMM15 (and shown in Figure 1) is from the European Marine Observation and Data Network (EMODnet Portal, September 2015 release). The increased resolution of both AMM15 and this EMODnet data set allows for improved representation of fine-scale features and processes, particularly along the shelf-break. The original EMODnet data

is referenced to Lowest Astronomical Tide (LAT), so has been converted to mean sea level (MSL) for use in the model. The differences between LAT and MSL referenced bathymetry are negligible in the deep ocean, but can be large on shelf, particularly in shallow coastal areas with large tidal ranges. Further details on the model domain and bathymetry can be found in Graham et al. (2018a).

Tidal modelling requires a non-linear free surface and this is facilitated in NEMO by using a variable volume layer method. The short time scales associated with tidal propagation and the free surface require a time splitting approach, splitting modes into barotropic and baroclinic components. The model uses a non-linear free surface, an energy and enstrophy conserving form of the momentum advection, and a free slip lateral momentum boundary condition. The tracer equation's use a TVD (Total Variance Diminishing) advection scheme (Zalesak, 1979).




**Table 1: Summary of the AMM15 –AMM7 model differences.**

| Model differences | AMM7 | AMM15 |
|---|---|---|
| Geographical domain | 40°N- 65°N<br>20°W – 13°E<br>Regular grid | ~ 45°N - 63°N<br>~ 20°W - 13°E<br>The grid has a rotated pole, chosen so that the *grid-equator* runs through the domain to reduce the distortion of cells with increasing latitude. While the rotated latitude is constant, the longitudinal grid steps range from ~1.47 km to 1.5 km. |
| Bathymetry | GEBCO corrected by NOOS partners | EMODnet 2015 |
| Horizontal resolution | 7km | 1.5km |
| Timestep | 300s (10s barotropic sub-timestep) | 60 s (~3.5s barotropic sub-timestep) |
| Penetrative radiation | 1-band shortwave radiation light attenuation (as used in POLCOMS, Holt and James 2001) | NEMO tri-band Red-Blue-Green (RGB) |
| Bottom friction | Log layer. Minimum drag coefficient $1.0 \times 10^{-3}$ | Log layer. Minimum drag coefficient $2.5 \times 10^{-3}$ |
| Momentum diffusion | bi-Laplacian on model levels $1 \times 10^{10}$ m$^4$/s | bi-Laplacian on model levels $6 \times 10^7$ m$^4$/s |
| Tracer diffusion | Laplacian on geopoential levels 50 m$^2$/s. | bi-Laplacian on model levels $1 \times 10^5$ m$^4$/s |

Since both AMM15 and AMM7 have a similar vertical grid, the vertical parameterizations remain similar. The generic length-scale scheme is used to calculate turbulent viscosities and diffusivities (Umlauf & Burchard, 2003). Dissipation under stable

5   stratification is limited using the Galperin limit of 0.267 (Holt & Umlauf, 2008). Bottom friction is controlled through a nonlinear log layer, with a minimum drag coefficient of $2.5 \times 10^{-3}$ (compared with a coefficient of $1.0 \times 10^{-3}$ in AMM7).

As many more fine-scale mixing processes are resolved in AMM15, only minimal eddy viscosity is applied in the lateral diffusion scheme. For momentum and tracers, bi-Laplacian viscosity is applied on model levels with coefficients of $6 \times 10^7$

10   and $1 \times 10^5$ m$^4$/s, respectively. For AMM7, additional viscosity and eddy diffusivity must be parameterized. A bi-Laplacian scheme is used on model levels for momentum, with a coefficient of $1 \times 10^{10}$ m$^4$/s. For tracer diffusion, a Laplacian diffusion scheme is used on geopotential surfaces, with a coefficient of 50 m$^2$/s.





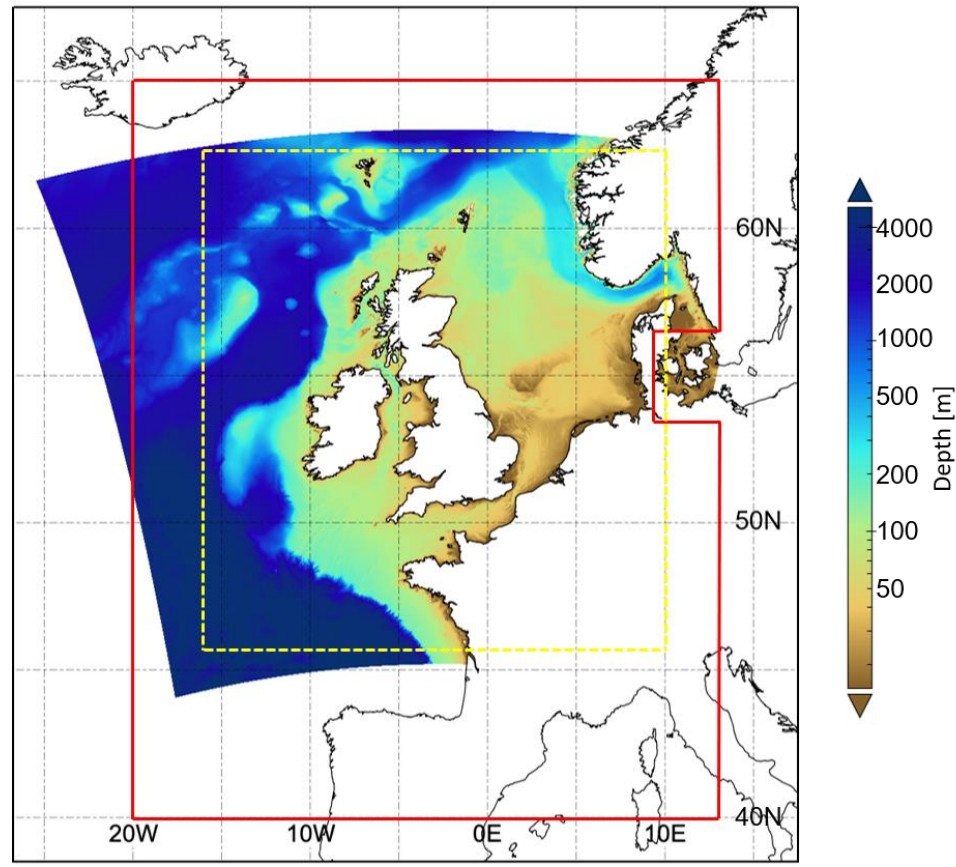

**Figure 1: EMODnet bathymetry, in meters (logarithmic scale), showing the AMM15 model domain. The red line defines the AMM7 model domain. The yellow dotted box is the domain covered by the AMM15 products delivered on a regular grid to the Copernicus users. Figure modified from Graham et al. (2018). The bathymetry colour range has logarithmic scale.**

### 2.1.1 Boundary and Surface Forcing

Tidal forcing is included both on the open boundary conditions via a Flather radiation boundary conditions (Flather, 1976) and through the inclusion of equilibrium tide. The Topex Poseidon cross-over solution (Egbert and Erofeeva, 2002; TPX07.2, Atlantic Ocean 2011-ATLAS) provides 12 constituents for amplitude and phase of surface height and velocity at 1/12°.

10  The model is one-way nested with the Met Office Operational North Atlantic 1/12° deep ocean model (Storkey et al., 2010) and the Copernicus operational Baltic Sea model (Berg et al., 2012). They provide temperature, salinity, sea surface height (not at the Baltic boundary), and depth integrated currents at the open boundaries. The two models, AMM7 and AMM15, are both nested in the open Atlantic and Baltic boundaries to the same products, but the boundaries are in a different geographical position due to the different model domains.

AMM7 and AMM15 are forced at the air-sea interface by two different Numerical Weather Prediction (NWP) outputs, the Met Office Unified Model (MetUM) global atmospheric model for AMM7 and the European Centre for Medium-Range Weather Forecasts (ECMWF) operational Integrated Forecasting System (IFS) for AMM15, see Table 2 for details. The Copernicus Marine Environment Monitoring Service requested the change from the MetUM to IFS forcing with the aim of

20  minimizing differences among the regional systems covering all the European seas. The fields from ECMWF and Met Office



have a similar spatial resolution but IFS fields for wind and atmospheric pressure have a lower temporal resolution as described in Table 2. The IFS analysis is available only at a low temporal resolution (6 hours) therefore the decision was made to force the system using forecast fields only (3 hourly), from the 00:00 UTC forecast base time run. A specific set of experiments are needed to assess the impact of this choice but are not within the scope of this paper. ECMWF products at higher temporal

resolution are now available and will be used in future releases of this operational system, improving the atmospheric forcing of this first version of AMM15. The IFS forcing is applied using the CORE (Common Ocean-ice Reference Experiment) bulk formulae (Large and Yeager, 2009). The specific humidity, sH, not available from the IFS field, is computed from the dew point temperature at 2m and the surface pressure using the World Meteorological Organization formulation (WMO, 2010):

$$sH = \frac{mwa * 100 * rSP * 10^2}{SP - (1.0 - mwa) * 100 * rSP * 10^2}$$

Where *mwa* is the ratio between the molecular weight of water and of dry air; *rSP* is the reference surface pressure; *SP* the surface pressure. The AMM7 instead is forced at the surface by direct fluxes from MetUM and using the COARE4 bulk formulae, as described in O'Dea et al. 2012.

An atmospheric pressure gradient force is applied at the surface of both models, using the atmospheric pressure field from MetUM and IFS respectively which affects the model free surface elevation.

The light attenuation in AMM15 is set to the standard NEMO tri-band scheme (RGB), assuming a constant Chlorophyll concentration (Graham et al. 2018a). AMM7 uses the single band light scheme previously used in the Proudman Oceanographic Laboratory Coastal Ocean Modelling System (POLCOMS) and outlined in Holt and James (2001). In this single band light scheme, the extinction depths vary across the domain in proportion to the bathymetry in order to estimate the change in water clarity between deep and shallow waters.

For AMM15, river run-off is based predominantly on a daily climatology of gauge data averaged for 1980–2014. UK data were processed from raw data provided by the Environment Agency, the Scottish Environment Protection Agency, the Rivers Agency (Northern Ireland), and the National River Flow Archive (gauge data were provided by Sonja M. van Leeuwen, CEFAS, Lowestoft, UK, personal communication, 2016). For major rivers that were missing from this data set (e.g. along the French and Norwegian coast), data have been provided from an earlier climatology (Young and Holt, 2007; Vorosmarty et al.,

1998), based on a daily climatology of gauge data averaged for the period 1950-2005, which is the climatology used by AMM7. The differences between AMM15 and AMM7 river discharge data are expected to be mainly, but not only, along the UK coastline.

**Table 2: AMM7 and AMM15 forcing description.**

| Forcing | AMM7 | AMM15 |
|---|---|---|
| **Surface forcing** | Met Office Global Unified Model (MetUM) Atmospheric model NWP analysis and forecast fields, calculated in the MetUM using COARE4 bulk formulae (Fairall et al. 2003). | ECMWF Integrated Forecasting System (IFS)-Atmospheric Model High Resolution (HRES) operational NWP forecast fields using CORE bulk formulae (Large and Yeager 2009) |
| **Surface forcing resolution** | Horizontal grid: ~10 km (2560 x 1920 grid points)<br>Frequency: 3 hourly mean fluxes of long and short wave radiation, moisture, 3 hourly mean air surface temperature but hourly 10m winds and surface pressure | Horizontal grid: ~14 km (0.125°x0.125°).<br>Frequency: 3 hourly instantaneous 2m dew point temperature, surface pressure, mean sea level pressure, and 2m air temperature. 3 hourly accumulated surface thermal and solar |

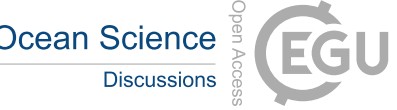

| | | radiation, total precipitation, and total snow fall. |
|---|---|---|
| **River run-off** | Daily climatology of gauge data averaged for 1950–2005. Climatology of daily discharge data for 279 rivers from the Global River Discharge Data Base (Vörösmarty et al., 2000) and from data prepared by the Centre for Ecology and Hydrology as used by (Young and Holt, 2007). | Daily climatology of gauge data averaged for 1980–2014. UK data were processed from raw data provided by the Environment Agency, the Scottish Environment Protection Agency, the Rivers Agency (Norther Ireland), and the National River Flow Archive (personal communication by Sonja M. van Leeuwen, CEFAS, 2016). For major rivers that were missing from this data set (e.g. along the French and Norwegian coast), data have been provided by the same climatology used by AMM7 (Vörösmarty et al., 2000 and Young and Holt, 2007). |
| **Lateral boundaries** | Met Office FOAM North Atlantic (1/12°; 6 hourly fields) and CMEMS Baltic Sea (2km, 1 hourly fields). | |
| | AMM7 and AMM15 have Atlantic and Baltic boundaries in a different geographical location. | |

### 2.2 Assimilation method

The data assimilation component of FOAM is NEMOVAR, a multivariate, multi-length scale assimilation scheme developed
collaboratively by the Met Office, the Centre Europeen de Recherce et de formation avancee en calcul scientifique, the ECWMF, and the Institut National de Recherche en Informatique et en Automatique (Mogensen et al., 2012). This has been implemented at the Met Office as an incremental 3D-Var, first guess at appropriate time (FGAT) scheme for the 1/4° global model (Waters et al., 2015) and the 7 km Atlantic Margin Model (AMM7, King et al. 2018).

An assimilation window of 24-hours is used, and observations assimilated include in situ and satellite swath SST observations, altimeter measurements of SLA (in regions with depth>700 m), and profile observations of the sub-surface temperature and salinity from a number of sources as detailed in Table 4. Increments are applied to the model fields at each time-step using the incremental analysis update procedure (IAU, Bloom et al. 1996). This Met Office implementation of NEMOVAR includes bias correction schemes for both SST and altimeter data (Lea et al. 2008).
In the AMM15 implementation described here the assimilation component has been upgraded to NEMOVARv4, which introduced a small number of changes compared to the scheme used in AMM7 (detailed in Table 3). In the AMM7 configuration, which uses NEMOVARv3, the SST bias correction scheme employs observations-of-bias (determined by matching nearby, contemporaneous in situ and satellite observations) to estimate a daily correction to the observations from each SST satellite. In NEMOVARv4 a variational bias correction has been introduced which combines information from
average SST innovations with the observations-of-bias used previously. We have also included observations from VIIRS in our reference dataset against which the other satellite SST observations are bias-corrected. This new scheme has been shown to be more resilient to changes in the observing system and gaps in observation coverage (While & Martin 2018). The SLA bias correction in AMM15 is unchanged from AMM7.





Although the same observation and background error variances are used in AMM15 as in AMM7 (see King et al. 2018), the background error correlation length-scales have been modified. The spatial covariance of background errors is modelled using an implicit diffusion operator with the horizontal length-scales specified a priori, and the vertical length-scales specified using a parameterisation based on the mixed-layer depth. In NEMOVARv3 this was modelled using three 1D diffusion operators,

5 but in NEMOVARv4 it is modelled using a 2D horizontal diffusion with a 1D vertical diffusion (Weaver et al. 2016). In both systems, two horizontal correlation length-scales are used: 100km for the long length-scale and the Rossby radius of deformation for the short length-scale. To avoid numerical computation issues, the short length-scale is restricted to have a minimum value equivalent to 3-times the grid-scale. This has the result that in shallow water the short length-scale for AMM15 can be as small as 4.5 km compared to 21 km for AMM7.

**Table 3: AMM15-AMM7 differences in the data assimilation scheme.**

| Data Assimilation | AMM7 | AMM15 |
|---|---|---|
| **NEMOVAR version** | V3 | V4 |
| **SST bias correction scheme:** | Offline observations-of-bias scheme. Reference dataset: in-situ. | Variational scheme in addition to observations-of-bias. Reference datasets: in-situ (drifters only) and VIIRS satellite data. |
| **Correlation operator short scale: 3-times grid scale** | ~20 km | ~5 km |

**Table 4: List of observations used for data assimilation.**

| Type | Fields | Platforms/Satellite | Data source |
|---|---|---|---|
| **IN SITU** | SST Temperature and salinity profiles | <ul><li>Ships</li><li>Drifters</li><li>Fixed moored arrays</li><li>Gliders</li><li>XBTs</li><li>CTDs</li><li>ARGO</li><li>Ferry boxes</li><li>Recopesca buoys*</li><li>Thermosalinograph</li></ul> | GTS; http://www.wmo.int/pages/prog/www/TEM/GTS<br><br>CMEMS-INS-TAC http://marine.copernicus.eu/ INSITU_GLO_NRT_OBSERVATIONS_013_030 |
| **SATELLITE** | SLA Along Track* *along with the corrections necessary for the use with a wind and pressure forced, tidal coastal model. SLA are assimilated only in deep regions (> 700m).* | <ul><li>Cryosat-2</li><li>Altika</li><li>Jason 3</li><li>Sentinel 3a</li></ul> | CMEMS SL TAC: http://marine.copernicus.eu/ SEALEVEL_EUR_PHY_ASSIM_L3_NRT_OBSERVATIONS_008_043 |
| | SST L2p/L3c | <ul><li>NOAA-AVHRR</li><li>MetOp-AVHRR</li><li>SEVIRI</li><li>VIIRSG</li><li>AMSR2</li></ul> | Group for High-Resolution Sea Surface Temperature (GHRSST) www.ghrsst.org |



### 2.3 Operational production

The FOAM system produces daily 2-day analyses (Best Estimate and NRT analysis) and a 6-day forecast (Figure 2). The timeliness of the observations, in situ and from satellite, significantly affects the number and the quality of the observations
available in the 24 hrs preceding the forecast, so two analysis cycles of 24 hr each are run to include as many observations as possible in the data assimilation.

The observations are downloaded from different sources: Copernicus Marine Environment Monitoring Service (CMEMS) for Sea Level Anomaly L3 products, Group for High Resolution Sea Surface Temperature for the L2 SST satellite observations
and the Global Telecommunication System (GTS) and CMEMS for the in-situ observations. The vertical profiles need to be thinned to reduce the spatial error correlation between the observations. For a given 24 hours, they are thinned in 3D space with the values: $\Delta$lon, $\Delta$lat=0.2°, $\Delta$z=10m. Prior to data assimilation, a quality check of the observation is performed using the model background produced by the forecast cycle of the previous day. Observations flagged as bad are not used for the data assimilation. The quality check for the vertical profiles is performed at 51 geopotential standard depth. The full profile is
rejected only if more than 1/3 is flagged as bad. The QC information, the background and observations fields are stored in specific files, known as feedback files. The error thresholds for the QC are set in order to avoid unrealistic model fields due to bad observations.

The lateral boundaries for the Atlantic region and the Baltic are both from the forecast production of the previous day. The
Baltic boundaries are downloaded every morning a few hours before starting the operational suite for the NWS. The CMEMS Baltic Sea product has 5-day forecast, produced twice a day, at 00:00 and at 12:00 and delivered at 07:00 and 19:00 UTC respectively. The NWS models are forced with the latest data available at 05:00 UTC and the last hourly field is persisted to produce the last day of forecast.

The atmospheric forcing is downloaded from ECMWF and since the last set of data is available at 07:00 UTC, production is started 15 minutes later to allow for download delays. Once the model and data assimilation task are over, the post-processing task starts because the products need to be organized for delivery to the users. The raw output files are interpolated on a standard grid at the same resolution as the model rotated grid, 1.5 km, but covering a slightly smaller area (see the yellow rectangular dotter line in Figure 1). The vertical terrain-following levels are converted for users' convenience, into 33 standard
geopotential levels: [0, 3, 5, 10, 15, 20, 25, 30, 40, 50, 60, 75, 100, 125, 150, 175, 200, 225, 250, 300, 350, 400, 450, 500, 550, 600, 750, 1000, 1500, 2000, 3000, 4000, 5000 m ]. All the files are then packaged to be compliant with the Copernicus and CF standards. Each day the best estimate [T-48h, T-24h] and NRT [T-24h, T+00h] analyses are delivered as well as 6 forecast days for all products (with the first day of the forecast being for the day of production). The delivered products are 25 hr daily means and hourly instantaneous products of temperature, salinity, horizontal currents, Sea Surface Height (SSH), Mixed Layer
Depth (MLD). Daily mean values are calculated as means of 25 instantaneous hourly values, starting at midnight and finishing on the following midnight to remove the tidal cycles. The data are in netCDF4 format and the volume of each production cycle is ~14 Gb (1.7Gb for each day), while for AMM7 is ~1Gb. The production process takes approximately 4 hours, four times that required by AMM7. It is planned to improve the robustness of this first implementation of the system by improving the dependencies of the different tasks in the operational suite and investigating ways of reducing the dependency on IFS delivery.
The quality of the products and the observations used for the production are monitored each day to allow the ocean forecasting scientists to take action if there are anomalies in the production or missing observations, and to allow users to be promptly alerted in the case of degradation of the products.





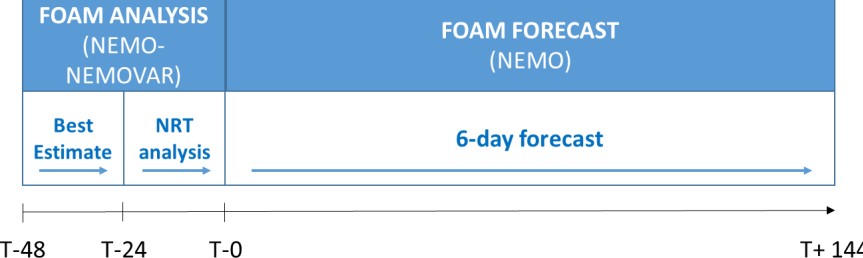

**Figure 2: FOAM production cycle**

### 3. Trial experiments

The assessment of the pre-operational implementation is based on trial experiments covering the years 2016-2017. The strategy

for trialling forecasting systems prior to entering into operations is one of significant debate at the time of writing. Ideally, given the relatively poor in situ observational coverage, a long period of trialling would be used to assess the system performance to gain a full (and statistically significant) understanding of its performance in all seasons and under a range of conditions. However, a combination of the length of time and computational cost to run those trials, the overhead in preparing observations and fluxes and the difficulty in finding a period of observations and fluxes that truly reflect the operational

conditions, lead to a more pragmatic approach being taken. It should be noted that the AMM15 modelling system itself has already been assessed for a long period and its quality documented in Graham et al., 2018a. Similarly, the data assimilation methodologies are well-tested (King et al., 2018) and have been robust in operations in other implementations. This assessment is therefore complementary and allows an assessment of the system as it is in operations, with fluxes, boundaries and assimilated observations used that are similar to the operational system. The trial experiments are required to cover a period

in the recent past in order to avoid differences in the observational network and/or in the forcing resolution/quality. The two years chosen are therefore a balance between covering a multi-year period, that is recent in time (and hence representative of the operational conditions) and achievable on the timescales required to transition into operations.

Before adding a new product to the operational production, the system must be shown to offer an improvement over the

previous system. For AMM15 this was done by setting up comparative trials running the existing and the new system, which are running with different forcing and initial conditions. This assures that we reproduce the operational products, new and old, and we validate the quality of the products.

The operational version of AMM7 was re-run rather than using the operational products produced in real time in order to avoid inconsistency in the number and type of observations assimilated by the two systems. Indeed, the real time production can

suffer temporary delays or problems in the delivery of the observations that are not reproducible in delayed time.

A free (non-assimilative) run was performed as a control, for both AMM7 and AMM15, but the results are not described in this work, since our aim is to assess the quality of the products delivered to the users. Apart from the resolution, the major differences between AMM7 and AMM15 are in the initial conditions, the atmospheric forcing and the location of the lateral

boundaries.

The AMM7 initial conditions are from the operational (assimilating) system while for AMM15 they are from an extension of the non-assimilating experiments presented in Graham et al. (2018a), which finished at the end of 2014. This run was extended to include 2015 and was run with the same source of atmospheric forcing and Atlantic lateral boundary used for 2016-2017.

However, the Copernicus Baltic datasets used to calculate the boundaries used in operations for AMM7 are no longer available



due to the Copernicus retention policy, and so cannot be used to calculate the AMM15 equivalents. We therefore used the Copernicus product where it was available (for the years 2016 and 2017) but for 2015 a General Estuary Transport Model (GETM, Burchard et al. 2002) implementation for the Baltic (as used by Graham et al. 2018a) was used in its place.

This paper details the assessment of the quality of the AMM15 operational system based on the assimilative run and is therefore representative of the analysis day. Graham et al. 2018a have demonstrated that the AMM15 without data assimilation performs equally or better than AMM7. A detailed assessment of the forecast based on the real time forecast produced since the beginning of November 2018 will be conducted in the future. It is anticipated that the forecast quality improvement for the AMM15 against the AMM7 will be greater than the improvement for the analysis day, given the improved underlying model,
but that must still be demonstrated.

**Table 5: Summary of the trial experiments.**

| System Name | Data Assimilated | Initial state | Forcing |
|---|---|---|---|
| **AMM7** | 3D SST, SLA, Sub-surface profiles | restart from AMM7 (assimilative) operational | direct fluxes- MetUM |
| **AMM15** | as above for the AMM7 | restart from an extension of Graham et al., (2018a) non-assimilating | bulk - IFS |

The results and validation of these trials are used for two purposes: as a basis for making a decision on whether to proceed
with the operational implementation, and to provide feedback to the researchers developing the models and data assimilation systems to prioritize their research activities.

The observations assimilated in the NWS system are SST from in situ and satellite; SLA from satellite; and sub-surface vertical profiles, as detailed in Table 4. The satellite measurements guarantee a good coverage of the area, especially for SST, while
the sub-surface profiles are variable in terms of number of observations and spatial/geographical distribution. Figure 3 shows the observations distribution during year 2016-2017 for sub-surface observations of temperature and salinity for the two most extreme seasons in terms of data distribution, winter (defined as December-January-February) and summer (June-July-August). In summer 2016 and 2017, there are very few observations on shelf, in particular in the North Sea and this has an impact on the quality of the assimilative runs. Compared to the trial experiments, done in delayed-time, the real-time analysis
can have a temporary decrease of quality due to timeliness issues affecting the real-time delivery of observations or poor quality-controlled data.



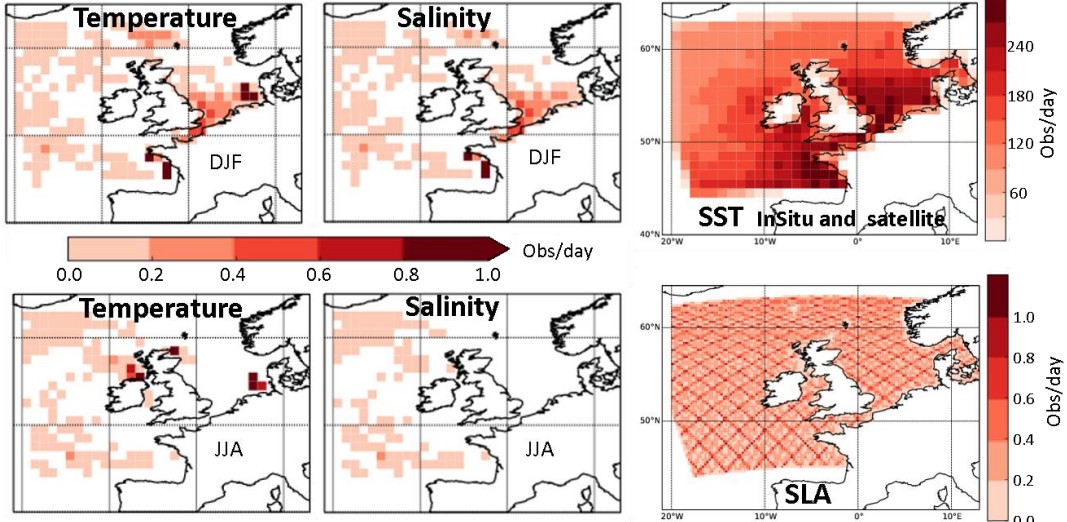

**Figure 3: Number of observations per day for various observation types over 2016/17: winter (DJF) temperature (top left) and salinity (top middle) profiles in 2-degree bins, summer (JJA) temperature (bottom left) and salinity (bottom middle) profiles, in situ and satellite SST (top right), and satellite SLA (bottom right).**

**4 Validation of the experiments**

Most of the observations used for the global validation are the same as used for the data assimilation, as described in the previous section. The model/observation differences are calculated before the model is corrected by the assimilation, but even so the observations are not fully independent from previously assimilated observations. All the same, this method is commonly used for model validation (King et al. 2018) and we consider the validation significant. Independent observations, available

on a very limited geographical domain and/or for a shorter period than the two years, have also been used, and have the benefit of providing an understanding of the impact of the high resolution locally on small areas and short time-scales. This approach differs from the validation described in Graham et al. 2018a, which focussed on the seasonal, interannual and multi-year time scales.

The independent observations we have used are:

•      Glider transects from the UK MASSMO4 experiments (north of Scotland);

          •      COSYNA HF radar in the German Bight

          •      Tide gauges

          •      Moorings in the German Bight and in the English Channel.

The geographical location of these observations is presented in Figure 4 where each type of observation is marked by a different colour. Also used was the Operational Sea Surface Temperature and Sea Ice Analysis (OSTIA) L4 SST product (Donlon, et al., 2012) which, although not assimilated, is not fully independent as it assimilates a similar (but not identical) set of SST observations as the AMM7 and AMM15 using similar methods (including using the same assimilation framework, albeit set up slightly differently).



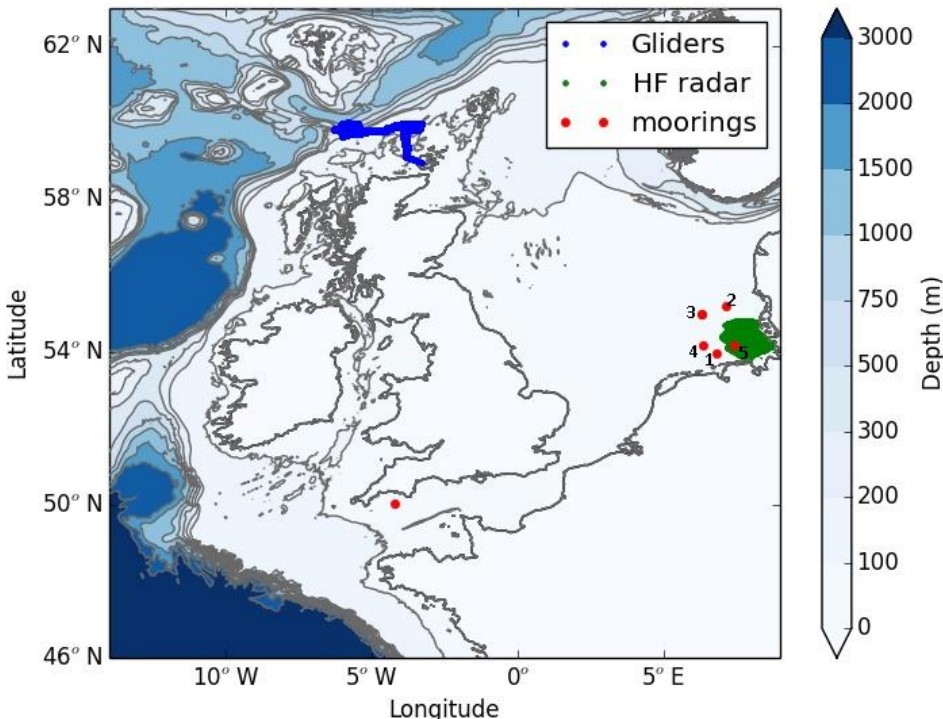

**Figure 4: Maps of the independent observations used for validation. MASSMO4 Gliders are blue, COSYNA HF in the German Bight in green and the mooring in the German Bight in red.**

The basin scale validation results for AMM7 and AMM15 are summarized in the Table 6 (with the system that has the best quality highlighted in bold), with a short description of the observations used. The ambition is to validate all the variables delivered to the users, even if there is a huge difference in the number and quality of observations available for the different parameters.

The RMS Difference and mean error (or bias) values are similar between the two systems and do not reflect the AMM15

system's superior performance as the validation at basin scale, averaged on the whole two years or one year period, penalises the high-resolution system. Whilst higher resolution models (subjectively) generate more realistic fields, it is often the case that statistics based on direct point match-up between interpolated model and observations do not improve due to the double penalty effect (e.g. Brassington, 2017). So, although global statistics do not show significant improvement from AMM7 to AMM15, it is demonstrated below that AMM15 consistently performs better than AMM7 when validated locally against high

resolution observations. It is an active area of research both with the ocean and Numerical Weather Prediction communities to understand how to quantitatively demonstrate skill improvement from higher resolution systems, and something that needs a real focus from our community.





**Table 6: Synthesis of the validation results for AMM7 and AMM15.**

| Variable | Location | Supporting observations | RMS Difference | | Mean Error (observation-model) | |
|---|---|---|---|---|---|---|
| | | | AMM7 | AMM15 | AMM7 | AMM15 |
| **M2 tidal harmonic (amplitude)** | Full domain | Tide gauge data | 10.4 cm | **9.8 cm** | **-0.2 cm** | -4.6 cm |
| **M2 tidal harmonic (phase)** | Full domain | Tide gauge data | 12.4° | **12.3°** | **-2 °** | 4.2° |
| **SST** | Full domain | In situ observations | **0.45°C** | 0.48°C | -0.01°C | -0.01°C |
| | Continental shelf | In situ observations | **0.51°C** | 0.54°C | -0.02°C | -0.02°C |
| **SST** | Full domain | OSTIA satellite L4 | 0.34°C | 0.34°C | **-0.06** | -0.08 |
| **T profiles** | Full domain | In situ observations | 0.47°C | **0.43°C** | -0.04°C | **0.02°C** |
| **S profiles** | Full domain | In situ observations | 0.13 PSU | 0.13 PSU | 0.01 PSU | -0.01 PSU |
| **SSH** | Off-shelf | Altimeter from satellite | 0.09 m | 0.09 m | -0.01m | 0.01m |
| | Continental shelf | Altimeter from satellite | 0.13 m | **0.11** m | -0.06 m | **-0.02** m |

**4.1 Tides**

Most of the continental shelf seas dynamics is dominated by tidal variability, which impacts the velocity fields and plays a key
role in the mixing and the generation of fronts. The improved resolution per se doesn't imply an improvement in capability to
model the tidal signal even if differences in the topography and coastlines could affect the baroclinic component of the tide
influenced by interaction of the flow with the bathymetry. This is particularly true in shallow areas when tidal currents are
strong. We have assessed the tides for year 2016 and 2017 using the tidal gauges. In addition, we used HF radar velocity data
available in a small part of the domain, in the German Bight (South-East North Sea), for a single month, March 2017.

**4.1.1 Tidal harmonics**

A harmonic analysis of the dominant tidal constituents was compared against tide gauge observations from BODC (British
Oceanographic Data Centre). The number of tide gauges taken into consideration for AMM15 and AMM7 is the same,
therefore the coastal buoys, not resolved by the AMM7 coastline, are not taken into consideration. AMM15 has a high
horizontal resolution but since the model applies a minimum depth of 10 m (same as AMM7), the inaccuracy in depth can still
affect its ability to properly estimate the tidal speed very close to the coast (Graham et al., 2018a). The statistics for the 7
dominant tidal constituents are detailed in Table 7. The differences in amplitude between AMM7 and AMM15 are small.
AMM15 has consistently lower RMSD for the phase of the tide, although the phase bias is similar or higher in AMM15.



**Table 7: RMSD and bias of the tidal amplitude and phase of the prevalent tidal constituents. The value are means over 292 tide gauges for both AMM7 and AMM15. The value in bold indicates an improvement.**

| Tidal Constituent | Amplitude (cm) | | | | Phase (deg) | | | |
|---|---|---|---|---|---|---|---|---|
| | RMS Difference | | Mean Error | | RMS Difference | | Mean Error | |
| | AMM7 | AMM15 | AMM7 | AMM15 | AMM7 | AMM15 | AMM7 | AMM15 |
| M2 | 10.4 | **9.8** | **-0.2** | -4.6 | 12.4 | **12.3** | **-2.0** | 4.2 |
| S2 | 4.1 | 4.1 | **-1.1** | -1.8 | 14.3 | **13.4** | **-3.5** | 5.5 |
| K1 | 1.7 | **1.6** | **-0.6** | -0.7 | 18.1 | **17.4** | **4.1** | 5.1 |
| O1 | 2.1 | **1.3** | 1.5 | **0.0** | 19.6 | **14.1** | **1.7** | 2.5 |
| N2 | 4.2 | **3.7** | -0.6 | **-0.3** | 31.0 | **26.7** | -4.4 | **2.6** |
| Q1 | 1.7 | **1.4** | -0.8 | **0.1** | 34.7 | **34.0** | -11.3 | **2.0** |
| M4 | 4.8 | **4.5** | **-1.0** | -1.9 | 89.6 | **66.8** | **-2.8** | 15.3 |

While the performance of AMM7 and AMM15 is similar (Table 7) for basin means, anomalies vary across the domain, showing regional improvements (Graham et al. 2018a).Figure 5 shows the spatial distribution of the M2 tidal errors in the two models. The values of RMS and mean error (model-observation) for amplitude and phase are very similar.

The M2 amplitude tends to be somewhat underestimated in the south-west part of the North Sea and along the east coast of
10 the UK. Phase errors, in both models, are largest in the southern North Sea and off the north-eastern coast of Northern Ireland. The AMM15 M2 amplitude is more accurate in the western part of the basin, in particular around the Kintyre peninsula as already described in Graham et al.(2018a), and in the Bristol channel area. The AMM15 phase error is smaller than AMM7 in the German Bight (South-East North Sea) but not in amplitude.

There are no significant differences in the co-tidal chart (not shown) between AMM7 and AMM15, both are very similar to
15 the charts shown in Graham et al. (2018a).





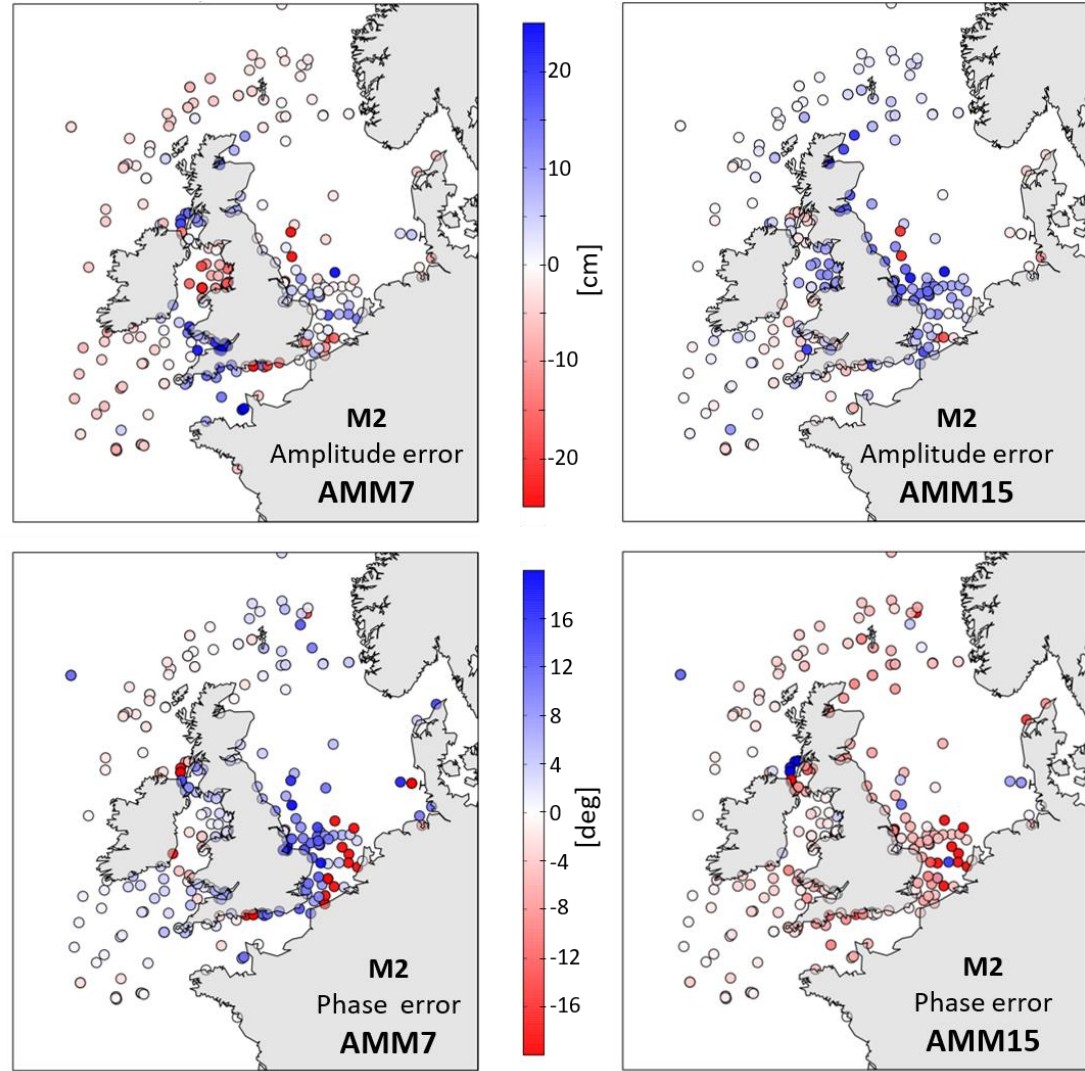

**Figure 5 : M2 Amplitude (top) and phase (bottom) error relative to observations (observations-model) for AMM7 (left) and AMM15 (right).**

**4.1.2 Tidal flow**

5    The HF radar data used here are for one month, March 2017, of total surface velocity estimated from 3 WERA HF radar systems deployed in the German Bight (Gurgel et al., 2011) as part of the COSYNA (Coastal Observing System for Northern and Arctic Seas) observing network (Figure 4). Data are available through the EMODnet Physics data portal. At the operating frequencies used, the total surface velocities represent an integrated velocity over a depth between 1 and 2 m. The HF radar data are averaged every 20 minutes on a grid of resolution of ~3 km.

10   Because the high frequency variability of the flow in the German bight is dominated by tidal flow, a low pass filter was used to separate the tidal and the residual component of the velocity. The tidal flow assessment presented here uses the high-pass filtered data (Figure 6); the residual currents that come from the low-pass filtered data are discussed in Section 4.5. Vector correlation (or complex correlation of $u+iv$, where $u$ and $v$ are the zonal and meridional velocity respectively, and $i$ is the





square root of -1) were estimated and are displayed as correlation amplitude, and phase, or veering. The phase represents the rotation between the two vectors that gives the highest correlation.

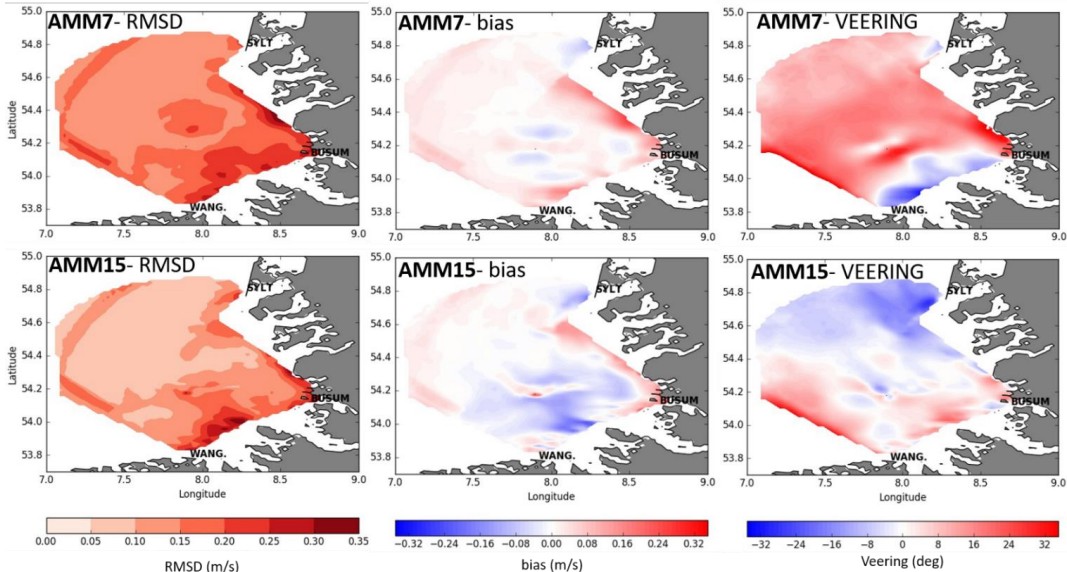

**Figure 6: plots of statistics between HF radar surface current observations and AMM7 (top) and AMM15 (bottom) for the high-passed filtered data (tidal signal). Bias (observation-model) and RMSD (in m/s) are estimated on the velocity vector magnitudes. Right panel: phase or veering. Positive (negative) veering represent a clockwise (counter-clockwise) angle of AMM7 (top) or AMM15(bottom) vectors with respect to the HF radar vectors.**

The various metrics show that AMM15 resolved the tidal current in the south German Bight better than AMM7, as shown in Figure 6. Although AMM15 bias is slightly higher than AMM7 in some areas, RMSD of AMM15 is smaller than the RMSD of AMM7 everywhere in the HF radar domain.

Both models show high correlation with the observations (not shown) and AMM15 has a lower phase, veering, with the observations. Because tidal velocities are rotating periodic signals, the spatial angular veering estimated using the complex correlation can also be interpreted as a temporal phase of the tidal signal. Positive veering angle means that the model tidal velocities lead the HF radar tidal velocities. Figure 6 shows that the tidal phase is improved in AMM15 compared to AMM7 in most of the domain, consistent with the results of the comparison with the tide gauges, 4.1.1.

### 4.2 Sea Surface Height

AMM15 and AMM7 SSH is assessed against the satellite Sea Level Anomaly data used for the data assimilation both on and off the shelf (Table 6), matching the model to the observation before the SSH data assimilation. The matchups are created by interpolating the model field to the observation location, at the model time step nearest to the time of the observations. It is worth noting that the assimilation of SLA is done only where the bathymetry is deeper than 700m, therefore no observations are assimilated on shelf and in a large part of the off-shelf region. The differences between AMM15 and AMM7 are negligible both on the continental shelf and off shelf. As expected in a tidally dominated area, on the continental shelf the RMSD is slightly higher than off shelf: 0.13 m on-shelf and 0.09 off-shelf. Both models are overestimating the SSH on shelf, but AMM15 has a smaller bias (-0.02m AMM15 and -0.06m AMM7). Instead, off-shelf AMM15 and AMM7 have the same absolute value of bias (0.01 m) but opposite sign, positive for AMM15 and negative for AMM7.



### 4.3 Sea Surface Temperature

Sea surface temperature is one of the key parameters of heat exchange at the air-sea boundary. Thanks to satellite and in-situ observations, SST is the variable with the best measurement coverage in our model domain. SST data is assimilated in the AMM15 and AMM7 models (Figure 4) during the Incremental Analysis Update (IAU) step of each model run. As a result, an

assessment of the model skill at predicting SST compared to observations would be expected to produce a positive result. We have compared the model against all the assimilated observations, the OSTIA products and a number of time series at selected moorings. It's worth to notice that while the comparison with the assimilated observations is done using the model output at the nearest time step, the comparison against OSTIA is done using the daily means. The hourly instantaneous fields are used instead for the comparison at the mooring locations. This validation allows us to have a general overview of the model SST

performance with a detailed analysis of the high-resolution model in few selected locations.

### 4.3.1 Comparison with in situ and satellite

Model SST has been assessed here against in-situ SST measurements matching the model to the observation before the data assimilation, at the model time step closer to the observations time. The in-situ measurements are from buoys and ships of opportunity. The number of the in-situ SST observations is pretty good during the two years: ~1000 obs/day on the full model

domain of which ~500 on-shelf. The differences between the two systems, in the full model domain, are very small (Table 6). The RMSD is ~0.5°C for both AMM7 and AMM15. Both models have a small warm bias, -0.01°C, over the full period. The warm bias is mainly due to the winter months when the model is slightly warmer than the observations. The same statistics on-shelf shows very similar results.

In addition to the in-situ observation assessment, the model hourly SSTs have been compared to the Met Office's Operational Sea surface Temperature and sea Ice Analysis (OSTIA) system (Donlon, et al., 2012). OSTIA provides analyses of the foundation SST (i.e. the SST free of diurnal variability) and assimilates in-situ and satellite observations. The OSTIA data, available through the Copernicus Marine catalogue, is produced on a 1/20°grid (~6km resolution), however, this is not the feature resolution of the product, which depends on other aspects of the system such as the correlation length scales used in

producing the analysis. OSTIA has a maximum feature resolution of ~20-30 km and so both AMM7 and AMM15 are expected to represent smaller features than OSTIA. The important point to consider in this assessment is that OSTIA foundation SST is being compared to the model surface box daily mean SST that should be biased warm compare to foundation.

The reference grid used for the inter-comparison of these three datasets is AMM7, therefore OSTIA and AMM15 have been interpolated at 7km.

The bias is defined as observation-model. Both models are biased warm compared to OSTIA (Figure 7), in agreement with the in-situ-model matchups statistics. AMM15 has a slightly higher bias than AMM7 but the same RMSD. The high variability of the signal in AMM15 could be penalized by the interpolation onto a grid at lower resolution. Overall, we see little difference in performance between the two systems. The mean RMSD for the period 2016-2017 is for both the systems of 0.3°C, smaller than the RMSD computed by the in-situ-model matchup statistics (0.5°C). The comparison between OSTIA-model and insitu-

model differs also because OSTIA comparison uses a full field to calculate the statistics rather than the single-point observation used in the insitu-model matchups.



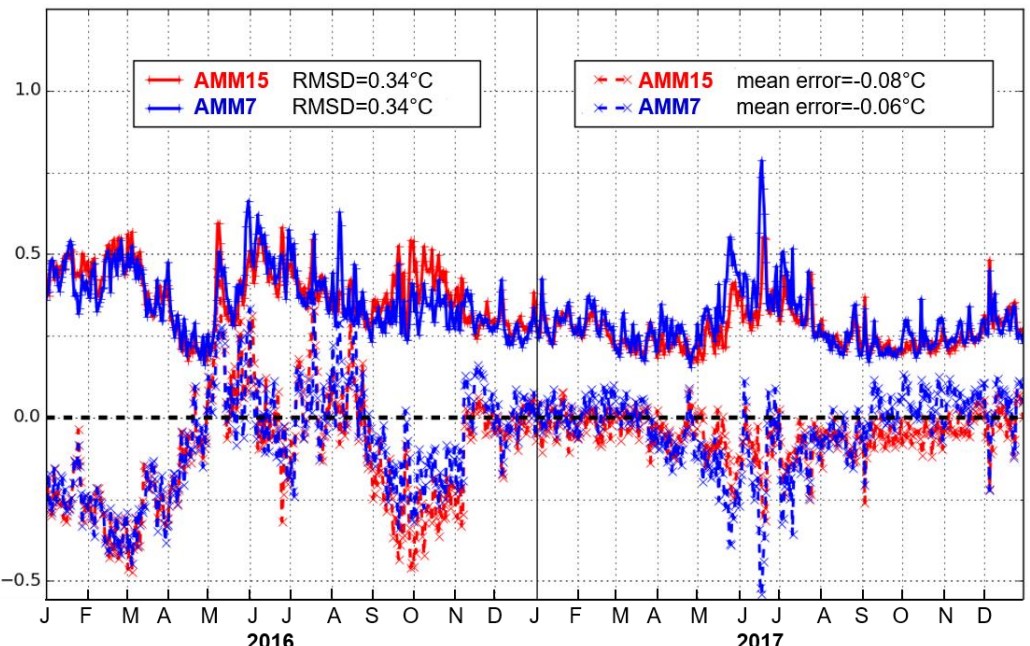

**Figure 7: OSTIA minus Model SST RMSD and bias daily comparison for AMM7 (blue) and AMM15 (red) for the whole domain (negative mean error indicates a warm model bias).**

### 4.3.2 Variability in SST

A small number of buoys have been used to investigate the SST variability in the models (Figure 4). Three sites were investigated in this study, E1 (50.026°N, 4.225°W) in the English Channel and the FINO sites in the German Bight (FINO1 at 54° 00.89 N, E 6°35.26 E and FINO3 at 55°11.7 N 7°9.5 E), buoys marked as 1 and 2 in Figure 4. In the future it would be helpful to get a broader range of sites included. A Butterworth filter has been applied to the hourly SST to remove low frequency (periods greater than 5 days) signals in the SST timeseries, to allow a comparison of the high frequency changes

only. The observation data were interpolated hourly to be equivalent to the model data, and the precision of the model data reduced to the same precision as the observations, to allow direct comparison and to prevent any aliasing. The timeseries were divided into seasons, both due to the high seasonal variability of SST and to avoid observation data gaps that would skew the analysis. We defined the four seasons as: December-January-February (DJF), March-April-May (MAM), June-July-August (JJA), and September, October, November (SON). The data used for this study covers the period December 2016-November

2017.

The model data was taken from the analysis day. It would be interesting to also assess the forecast, this will be the subject of future studies. Figure 8 shows the SST and filtered SST timeseries at the FINO3 mooring for two different seasons, winter (DJF) and summer (JJA). The filtered SST signal has a higher variability in summer when the diurnal warming is stronger and

20 therefore the SST gradients are bigger. AMM7 and AMM15 have very similar values of SST, probably due to the data assimilation of SST that brings both models close to the observations.





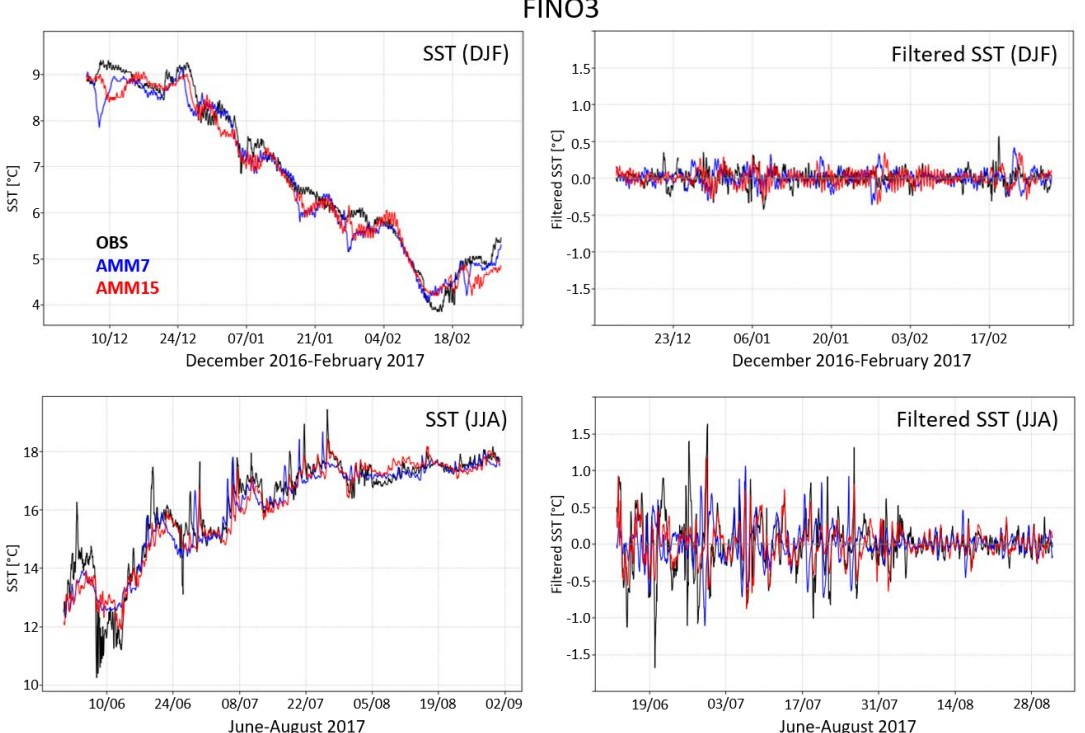

**Figure 8: Timeseries of sea surface temperature (left) and filtered SST (right) for the FINO 3 buoy for December 2016-February 2017 (top) and June-August 2017 (bottom). Observations are shown in black, AMM7 in blue and AMM15 in red.**

Spectral powers were estimated and smoothed using a Loess filter to remove noise in the spectra. Figure 9 shows the power

5 spectra for each season at the FINO 3 buoy, the non-filtered spectra are also plotted as faint lines. The power spectra of SST at the other mooring locations, E1 and FINO1, are not shown but are similar to FINO3. Although this is not exclusively true, the general trend is for the models to drop off in power more quickly with frequencies, and have a steeper spectral slope, than the observations. AMM15 SST is more variable at frequency higher than daily, although at periods of 4 hours and lower the models tend to behave quite similarly. This is consistent with what one would expect from the mesoscale resolving skills of

10 AMM15. This high frequency increase of variability in AMM15 compared to AMM7 can also be seen by quantitative inspection of the model fields (Figure 8), with small length scale features being more prevalent in AMM15. At higher frequencies (shorter periods) SST spectra for both models collapse to the same power spectrum value with generally less variability than the observations at high frequencies.



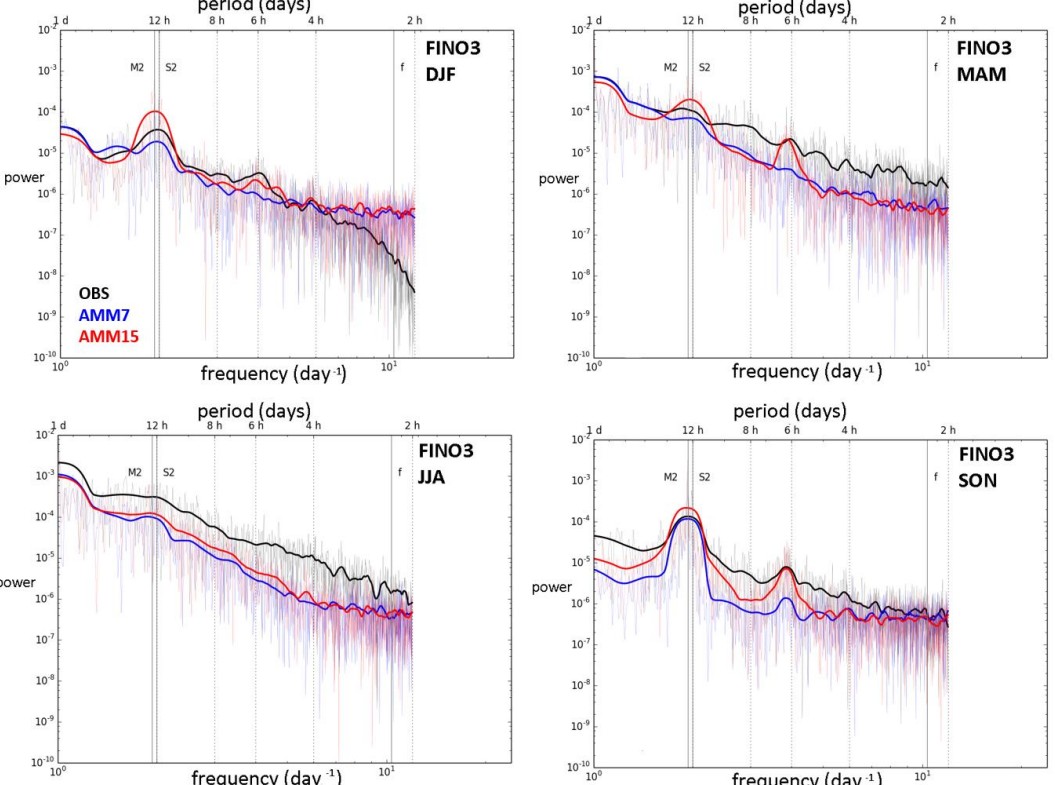

**Figure 9: Power spectrum of SST for the FINO3 buoy (black line) compared with the AMM7 (blue) and AMM15 (red) simulations for December 2016 to February 2017 (top left), March to May 2017 (top right), June to August 2017 (bottom left) and September to November 2017 (bottom right).**

This suggests that on average some of the very high frequency / small scale features are still not being represented in the models, although it should be noted that the model represents a mean over a grid which will by definition introduce some smoothing, whereas the observations are (at least to a greater extent) sampling at a point. This analysis demonstrates quantitatively that the AMM15 better represents the high frequency / small scale features, which can visually be observed from model fields time series, but are poorly assessed through global statistics, as shown in Table 6. This result is likely to be even

more pronounced in forecast fields, although that is not demonstrated here.

### 4.4 Water column

### 4.4.1 Temperature and salinity profiles

AMM15 shows improved vertical structure of the water column, with a lower bias and RMSD compared to AMM7 in salinity

and temperature. Figure 10 shows the mean error (obs-model) and the RMSD averaged over the whole domain, with observation-model differences calculated before the assimilation of the vertical profiles. The temperature bias of both models is very small at the surface but increases below 100m. Between 500 and 1000m AMM15 has a mean error close to zero, while AMM7 has a cold bias. AMM15 also has a lower RMSD than AMM7 at all depths below the surface. The distribution of the sub-surface observations, shown in Figure 3, is uneven, with very few observations on shelf, therefore it is not possible to

distinguish between the water column improvements off-shelf and on-shelf using this technique, and these improvements are demonstrated predominantly for the off-shelf region. This is a very good result considering that the skills in modelling salinity and temperature depth structure have recently been significantly improved in AMM7 with the improvements in data





assimilation, with the addition to SST of SLA and sub-surface profiles (King et al. 2018). This means that, in less than 2 years, the NWS system has consistently improved its skill in resolving the vertical profiles of temperature and salinity and therefore the density of the water column.

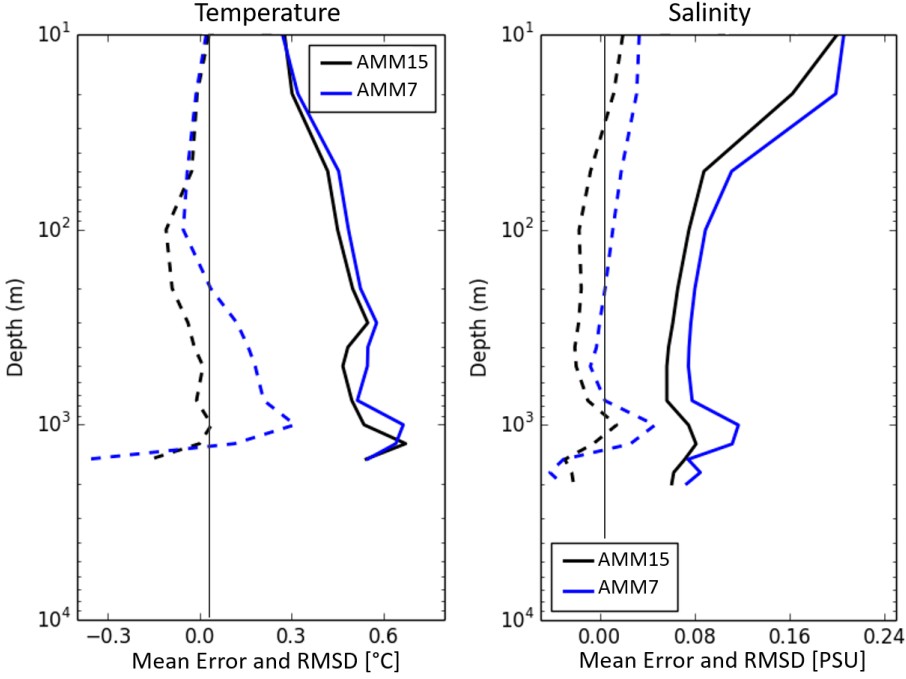

**Figure 10: Observation minus model temperature (left) and salinity (right) profile assessment for AMM15 (black) and AMM7 (blue) for the whole domain. The RMS difference is shown by the solid lines and mean error (observation-model) is shown by the dashed lines.**

### 4.4.2 Moorings in the German Bight

Table 8 and Table 9 show the temperature and salinity statistics for the model and observations comparison during year 2017 at the five moorings in the German Bight (Figure 4). Observations for both temperature and salinity are available at surface (5m) and bottom. The models assimilate only SST observations, not salinity and no bottom observations. The hourly instantaneous model data are compared to the buoy observations with a few discontinuities due to missing measurements during a short period of the year. AMM15 has a better RMSD and a lower error mean at all buoy locations. The high frequency

variability is better reproduced by AMM15 than AMM7, as shown in Figure 11 for the surface salinity field in the NsbII (mooring number 3 in the map of Figure 4).

Temperature RMSD and bias are very small at surface, due to the strong constraint of the data assimilation of SST (as described in 4.3) while at the bottom AMM15 is more accurate in prescribing the temperature at all mooring locations (Table 8).

AMM7 and AMM15 both have high salinity errors in the German Bight, as highlighted by the comparison with the buoys that are located closer to the coast (Fino1, Fino3 and UFSDeBucht). This is most probably due to representation of river discharge. AMM15 performs better than AMM7, probably because it is less diffusive within river plumes and has a lower lateral diffusion. Improved bathymetry and coastal resolution are also likely to play a role in coastal areas with depth less than 20m. AMM15 has halved the salinity error compared to AMM7 when compared with the outer buoys (NsbII and TWEms). It is encouraging

to see that AMM15 is better than AMM7 at the bottom at all mooring locations. The decision to use the climatological river



discharge dataset instead of E-Hype for AMM7, and subsequently AMM15, has improved salinity remarkably in the German Bight, reducing the model fresh bias. This modification was implemented in April 2017, meaning that we have significantly improved the salinity in the last two major updates of the NWS forecasting system. Nevertheless, using a climatological river runoff dataset is a limitation for a high-resolution forecasting system, affecting variability in coastal water properties. Finding

5   a suitable alternative will be a priority for future releases of this system.

Temperature RMSD and bias are very small at the surface, due to the strong constraints of the data assimilation of SST (as described in 4.3) while at the bottom AMM15 is more accurate in prescribing the temperature at all mooring locations (Table 8).

**Table 8: Yearly mean (2017) RMSD and bias statistics at the 5 moorings in the German Bight (observation-model). Surface and**
10   **bottom temperature for AMM7 and AMM15.**

| Buoy | Temperature (C°) | | | | | | | |
|---|---|---|---|---|---|---|---|---|
| | Surface | | | | Bottom | | | |
| | RMS Difference | | Mean Errors | | RMS Difference | | Mean Error | |
| | AMM7 | AMM15 | AMM7 | AMM15 | AMM7 | AMM15 | AMM7 | AMM15 |
| 1 Fino1 | 0.32 | **0.21** | **0.03** | -0.05 | 0.31 | **0.21** | 0.07 | **-0.03** |
| 2 Fino3 | 0.38 | **0.37** | **-0.02** | -0.04 | 0.96 | **0.59** | -0.38 | **-0.24** |
| 3 NsbII | 0.30 | **0.25** | 0.12 | 0.12 | 0.59 | **0.49** | **-0.13** | -0.14 |
| 4 TWEms | 0.28 | **0.26** | 0.13 | **-0.02** | 0.28 | **0.16** | 0.11 | **0.00** |
| 5 UFSDeBucht | 0.50 | 0.50 | 0.10 | **0.01** | 0.95 | **0.75** | **-0.31** | -0.33 |
| Mean value | 0.36 | **0.32** | 0.07 | **0** | 0.62 | **0.44** | **-0.13** | -0.15 |

**Table 9: Yearly mean (2017) RMSD and bias statistics at the 5 moorings in the German Bight (observation-model). Surface and bottom salinity for AMM7 and AMM15.**

| Buoy | Salinity (PSU) | | | | | | | |
|---|---|---|---|---|---|---|---|---|
| | Surface | | | | Bottom | | | |
| | RMS Difference | | Mean error | | RMS Difference | | Mean Error | |
| | AMM7 | AMM15 | AMM7 | AMM15 | AMM7 | AMM15 | AMM7 | AMM15 |
| 1 Fino1 | 1.17 | **1.02** | 0.97 | 0.97 | 1.10 | **1.02** | 0.95 | **0.95** |
| 2 Fino3 | 1.06 | **0.73** | **0.35** | 0.48 | 0.90 | **0.62** | 0.53 | **0.38** |
| 3 NsbII | 0.33 | **0.22** | 0.20 | **0.03** | 0.37 | **0.17** | 0.26 | **0.03** |
| 4 TWEms | 1.05 | **0.51** | 0.85 | **0.29** | 1.08 | **0.45** | 0.89 | **0.26** |
| 5 UFSDeBucht | **0.99** | 1.07 | **0.55** | 0.87 | 1.08 | **1.02** | **0.86** | 0.90 |
| Mean value | 0.92 | **0.71** | 0.58 | **0.53** | 0.91 | **0.66** | 0.70 | **0.51** |



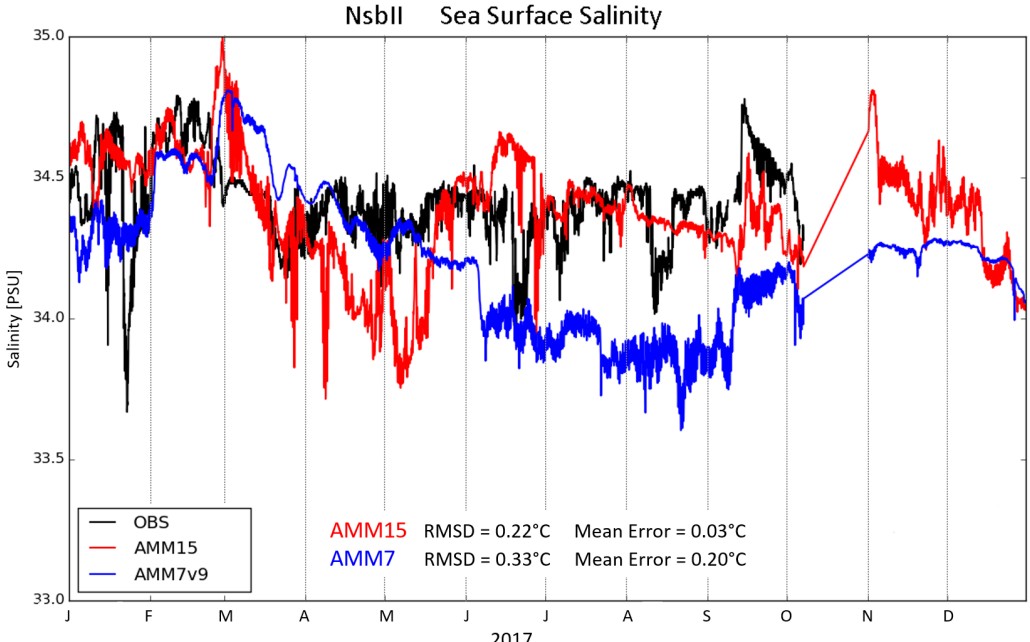

**Figure 11: Sea surface salinity at the NsbII mooring for year 2017. AMM15 red line, AMM7 blue line, observations black line.**

### 4.4.3 Glider transects

AMM15 and AMM7 vertical structure and high frequency variability is assessed against the glider profiles from MASSMO (Marine Autonomous Systems in Support of Marine Observations), Mission4 (Figure 4 and Figure 12). MASSMO is a pioneering multi-partner series of trials and demonstrator missions that aim to explore the UK seas using a fleet of innovative marine robots. With newly developed unmanned surface vehicles (USVs) and submarine gliders, the multi-phase project has successfully completed the largest single deployment of marine autonomous systems ever seen in the UK. In the summer of

2017 a fleet of 11 autonomous marine robots was deployed to explore the seas northwest of the Orkney Islands in search of marine mammals and sources of man-made noise pollution. The mission was part of an annual series of marine robot trials co-ordinated by the National Oceanography Centre in partnership with 16 organisations representing UK government, research and industry (http://projects.noc.ac.uk/massmo).The fleet comprised eight submarine gliders and three unmanned surface vehicles, travelling up to 200km offshore to the Faroe-Shetland Channel where water depths exceeded 1000m. The MASSMO4

campaign covered the period 22nd May to 6th June 2017 with 3 gliders deployed north of Scotland, close to the coast, and then travelled across the shelf break (Figure 4).

The MASSMO4 dataset is therefore a very high-resolution source of information in a key area of the model domain. We have compared the models and observations along the glider track, using the model high frequency data (hourly instantaneous fields). Figure 12 shows the trajectory of one of these gliders (553), which was measuring temperature and salinity from surface

to bottom. The background field in this figure shows surface salinity from AMM15 at 12:00 UTC of 23$^{rd}$ May, when the glider was in the position marked by the red dot.

AMM15 is in very good agreement with the observations and shows improvement, compared to AMM7(Figure 13 and Figure 14), particularly for salinity along the glider trajectories. The only exception being the low salinity pattern in the whole water column measured by the glider around the 23rd of March, when AMM15 is too salty and AMM7 too fresh. It could be due to




a misplaced of a front, as suggested by the AMM15 salinity map (Figure 12). The AMM7 salinity field (not shown) has lower variability and this could justify the smaller misfit compare to the glider in that precise location.

The salinity field in AMM15 has finer scale structures and usually the low salinity is better constrained along the coast. The density of the water column during the period of the gliders campaign is therefore much more accurate in AMM15 (Figure

15). While these results may not be representative of the whole model domain or of the seasonal variability in stratification of the water column, they are very encouraging.

In all depth profiles for AMM7, there is a white patch close to the bottom on the 23/05. This is due to the model bathymetry being shallower than the reality in that specific location. This is a confirmation than AMM15 has a more realistic representation of the bottom topography, as described in Section 2.1 Core Model Description.

The increased resolution of fine-scale structures in AMM15 results in increased transport across the shelf-break, particularly in the region observed here (Graham et al 2018b). These results, showing AMM15 has improved vertical structure and variability of the water column, support the conclusions of Graham et al. (2018b). Shelf-break processes transporting water masses between the deep ocean and across the shelf, will have a strong impact on conditions observed in this region.

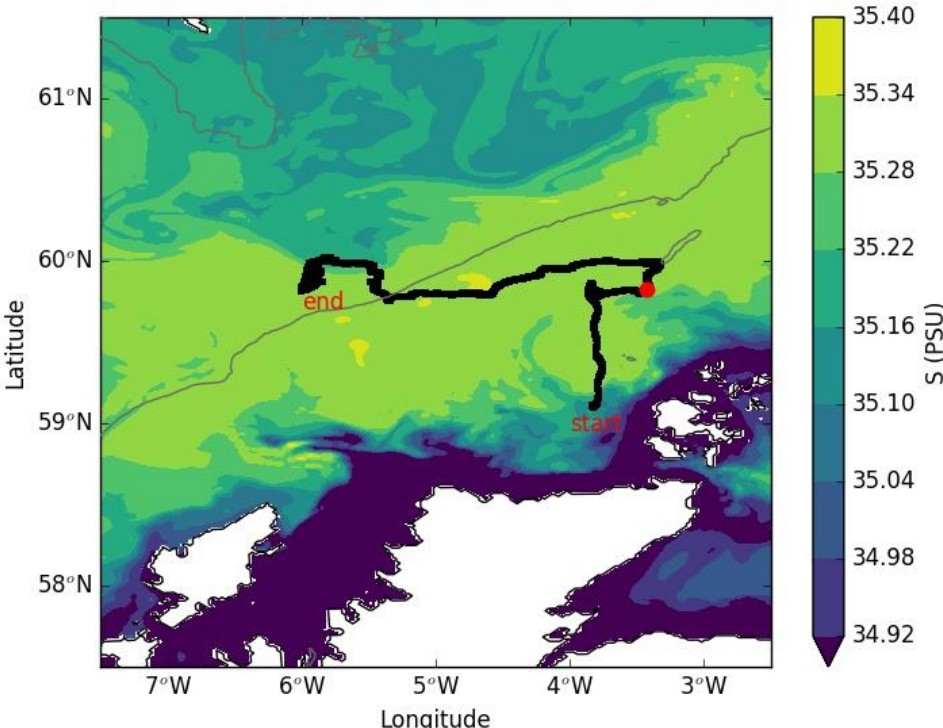

**Figure 12: Glider 553 trajectory. The glider started the measurement close to the Scotland coast, and then moved towards the shelf break. The black line is the glider trajectory. The grey line represents the 200m isobath. The red dot is the glider position on the 23rt May 2017. The field on the background in the salinity on the 23rd of May.**



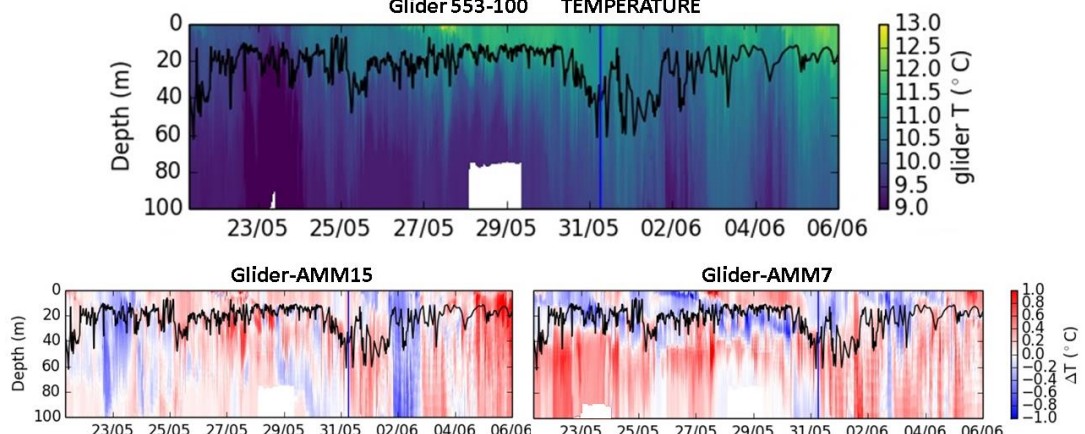

**Figure 13: Top 100m of temperature profile along glider 553-100 trajectory. Top panel: temperature measured by the Glider. Bottom left: difference glider-AMM15. Bottom right: difference glider-AMM7.The vertical blue line represents when the glider goes over the shelf break, crossing the 200m isobath. Before that time the glider is on the shelf. The black line is the MLD computed from the glider measurements.**

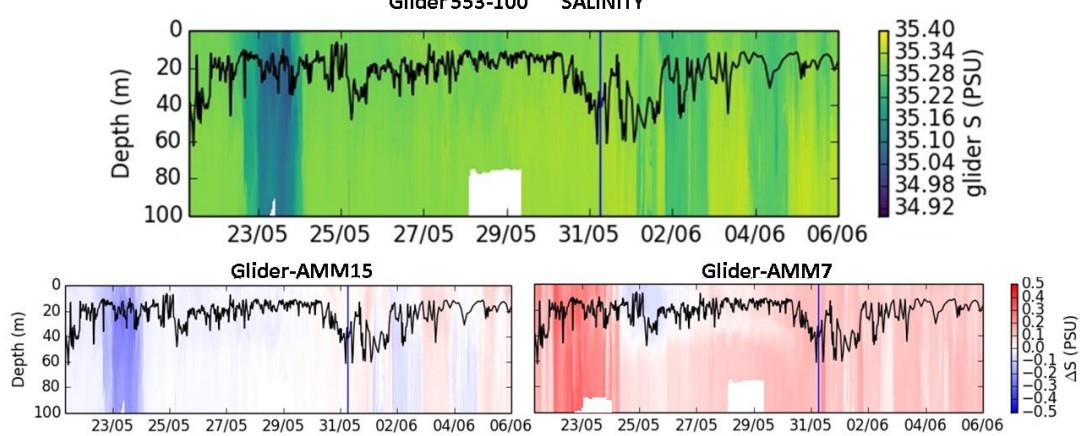

**Figure 14: Top 100m of salinity profile along glider 553-100 trajectory. Top panel: salinity measured by the Glider. Bottom left: difference glider-AMM15. Bottom right: difference glider-AMM7.The vertical blue line represents when the glider goes over the shelf break, crossing the 200m isobath. Before that time the glider is on the shelf. The black line is the MLD computed from the glider measurements.**





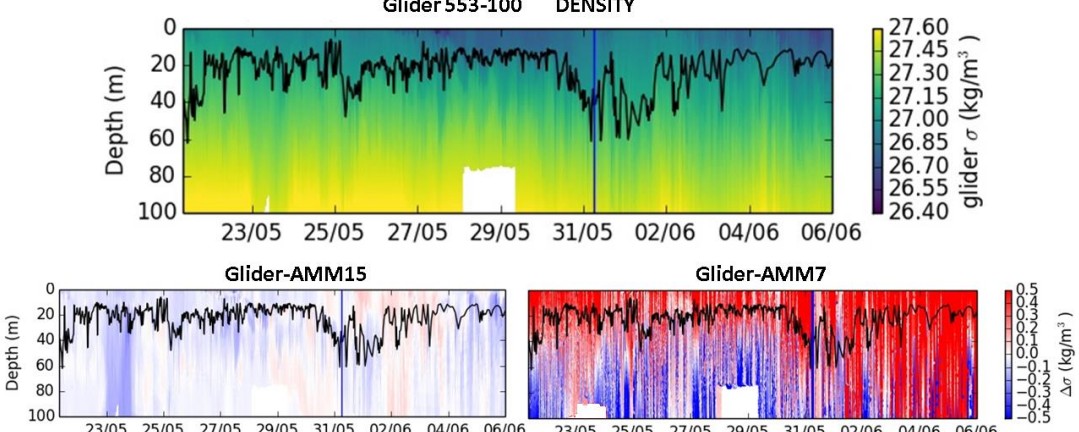

**Figure 15: Top 100m density profile along glider 553-100 trajectory. Top panel: salinity measured by the Glider. Bottom left: difference glider-AMM15. Bottom right: difference glider-AMM7.The vertical blue line represents when the glider goes over the shelf break, crossing the 200m isobath. Before that time the glider is on the shelf. The black line is the MLD computed from the glider measurements.**

### 4.4.4 Mixed layer depth

The mixed layer depth, MLD, has been calculated within the model using a density criterion, following the definition of Kara et al. (2000) except the reference depth of 10m is changed to 3m, due to the shallower regions of the continental shelf. The MLD is defined as the depth where the density increases, compared to density at 3m depth, corresponds to a temperature decrease of 0.2°C in local surface conditions. The EN4 (Good et al 2013) profile dataset of temperature and salinity was used to calculate an 'observed' Kara mixed layer depth following the same procedure used within the model.

An important point to note here is that on daily/monthly timescales the EN4 dataset is still relatively sparse in the region of interest and often clustered in particular locations. This is particularly true on the continental shelf. Assessing the model as a whole we see a seasonal cycle of errors with small errors in summer/autumn (bias ~10m and RMSD < 25m) and larger errors in winter/spring.

We have therefore also computed the mixed layer depth from the MASSMO4 observations, comparing the mixed layer depth from the gliders (black line Figure 15) with AMM15 and AMM7 in the corresponding locations (not shown). AMM15 reproduces the mixed layer depth better than AMM7 and represents the variability of the signal very well. This positive result was also true along the other glider trajectories in this region. AMM15, but not AMM7, also reproduces a deepening in the MLD at the shelf-break. While this could be just a temporary feature, it could also be explained by increased mixing due to internal waves, which begin to be resolved in AMM15, but not AMM7 (Guihou et al. 2017). Graham et al (2018b) also show that the slope current differs between AMM15 than AMM7 in this region, which is likely to affect the water column structure and variability around the shelf-break. Differences in currents are also discussed further in the following section.

### 4.5 Currents

A good forecast of the intensity and direction of the currents is needed for operations at sea, but the validation of this variable is particularly difficult due to the scarcity of observations. There are very few measurements of velocity in the model domain. Among the few data available we have decided to use the surface currents measured by the HF radar. The HF radar data used here are one month of total surface velocity currents estimated from 3 WERA HF radar systems deployed in the German Bight (Gurgel et al 2011). Data are available through the EMODnet Physics data portal. At the




operating frequencies used, the total surface velocities represent an integrated velocity over a depth between 1 and 2m. The HF radar data are averaged every 20 minutes on a grid of resolution of ~3km.

Because the high frequency variability of the flow in the German bight is dominated by tidal flow, a low pass filter was used to separate the tidal and the residual component of the velocity. Statistics were computed on the high-pass and low-pass

5    filtered velocity components to assess the tidal and residual flow respectively. The assessment of the tidal flow is described in section *4.1.2. Tidal Flow*.

AMM15 shows a clear improvement, with better defined mean residual surface velocity. In particular, the spatial variability observed in HF radar is better captured by AMM15 than AMM7 (Figure 16). Similar statistics as those estimated for the tidal flow (Figure 6) also show improvement in both amplitude and direction of the residual flow (not shown).

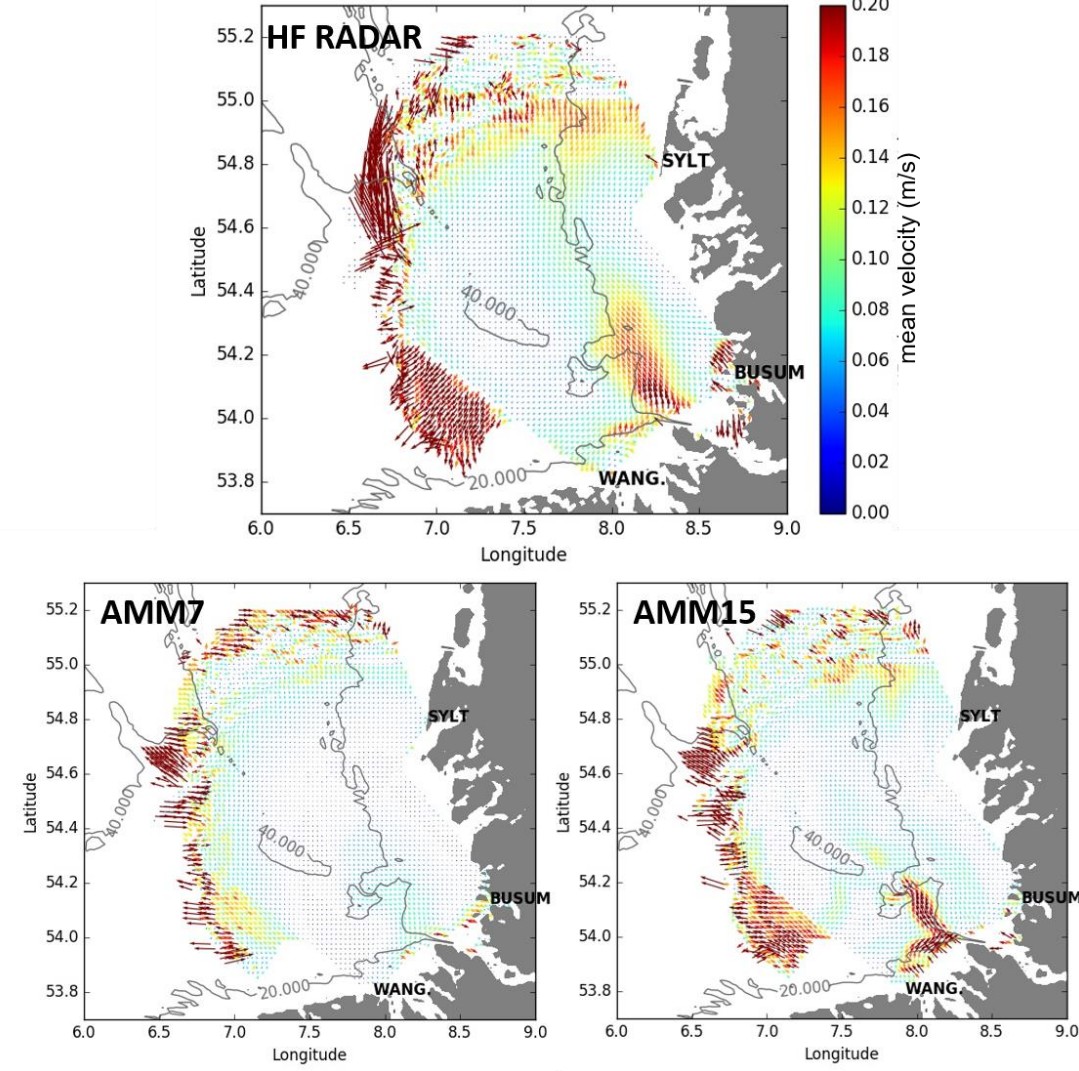

**Figure 16: March 2017 average surface flow for HF radar (top), AMM7 (bottom left) and AMM15 (bottom right). Model data are re-gridded on the HF radar grid using cubic interpolation. The hourly model output is linearly interpolated every minute to match the HF radar observations. Model output are only plotted where HF radar data are available.**

The current field of AMM15 is more detailed and seems more realistic than AMM7 but the lack of the observations makes it

15    difficult to properly assess the horizontal velocity field in key areas. Figure 17 represents the surface currents from AMM7

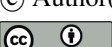



and AMM15 for a single day, to give an example of difference between the two models. AMM15 has a more complex current circulation in the deep part of shelf, with meanders and eddies, not resolved by AMM7. The European slope current, green arrow in Figure 17, is transporting Atlantic water into the North Sea, mainly through the Faroe-Shetland Channel (Marsh at al. 2017), influencing the characteristic of the water in the Northern North Sea and its circulation. The AMM15 current patterns

seems more realistic, reproducing eddies and meanders, has shown by drifters measurements done in this area (Burrows et al. 1999). The European slope current plays a major role in the across shelf transport, with AMM15 better reproducing the water exchanges as described in Graham et al 20018b.

The two models are also very different in the area characterised by the Norwegian Coastal Current (NCC) which is highlighted by the yellow vectors in Figure 17.

The NCC is a coastal current flowing from Skagerrak to the Barents Sea, following the Norwegian coastline, in the upper layer (50-100m) over the Norwegian Trench. This current is characterized by a front between low salinity water coming from the brackish Baltic sea and Norwegian coastal water, and the Atlantic water. This current has mesoscale meanders and eddies (Ikeda et al. 1989) propagating northward along the Norwegian coast. AMM15 reproduces better the mesoscale eddies and meanders of the NCC which are not resolved by AMM7.

The Scottish coastal current, seems to be well resolved by AMM15, with a strong current flowing along the west Scottish coast and meandering before entering the channel between mainland Scotland and the Hebrides (the Minch). This is a persistent current that interacts with the island chain of the Hebrides (Simpson 1986). AMM15 has a more detailed coastline and bathymetry in this area which is likely to be one of the main reasons why the model resolves this current. In AMM7, this current is almost absent, or too weak and misplaced to the west, instead of being, as it in AMM15, between the islands and the

mainland (Figure 17, red arrows). The contrast between the low salinity along the coast and the higher salinity of the Atlantic water is another key driver of this current (Simpson 1986), and this is better represented by AMM15, which has an improved salinity field compared to AMM7. AMM15 seems to be much less diffusive in the proximity of the river plumes keeping a narrower plume close to the coast and has a lower lateral diffusion (Graham et al. 2018a). AMM7 has low salinity water (less than 34.5 PSU) spreading much further away from the coast, all the way to the Outer Hebrides, while the salinity gradient in

AMM15 is located between the Minch and the western side of the Outer Hebrides. This assessment shows that AMM15 is in very good agreement with literature in this area. Further studies, and possibly targeted observations, are needed to validate this preliminary result and to assess AMM15 skills in predicting the seasonal variability of this current and the other currents described in this study.




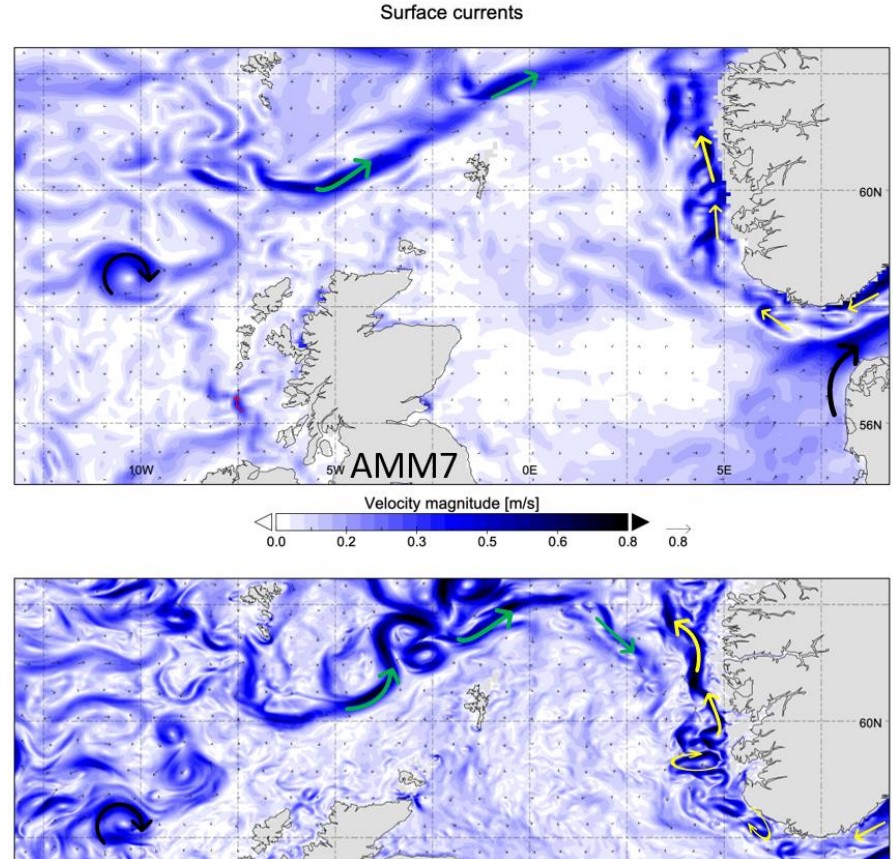

Figure 17: Surface current from AMM7 (top) and AMM15 (bottom), daily mean 02-12-2018.

**5 Conclusions and future developments**

The validation of pre-operational trial experiments for a new 1.5km resolution model of the European North West Shelf,
against observations and the predecessor 7km system, shows positive results.

AMM15 has improved skill compared to AMM7 and has proven to be an improvement especially when compared to high
spatial/timporal resolution observations. The currents pattern and variability are better reproduced by the new system, with
improved temperature and salinity throughout the whole water column. The most outstanding improvement seems to be in the
salinity which is closer to observations at basin scale and locally. Probably there are different factors which contribute to this
improvement. Firstly, salinity will be impacted by river runoff. While the two models have a similar daily climatological river
runoff data set it could differ locally. Despite similar runoff, the path of the river plume may also differ in the two models, so
may lead to local changes in salinity, for example in the German Bight, where the plume stays close to the coast rather than
diffusing off-shore. Secondly, the Atlantic and Baltic boundaries are in a different geographical location and this could imply
differences in the fluxes at the boundary. There is a strong salinity variability at the Baltic boundary and this can strongly





influence the salinity field and variability in the North Sea. There are ongoing developments to improve the Baltic boundary implementation in AMM15 that will help to understand further the impact of this boundary on the NWS. The Atlantic boundaries influence the exchange across the shelf and they could be partly responsible for improvements like those shown in the north of Scotland where the model has been compared with glider data and the AMM15 salinity field is much more realistic

than AMM7. The significant improvement in this area could also be due to AMM15 better resolving the flow through the Faroe-Shetland Channel and shelf-break exchange. The work from Graham et al. 2018b, shows that there is an increased flux across the shelf-break in AMM15 compared to AMM7. This could affect the exchange of water masses around the shelf break, and therefore influence salinity on the shelf. Another reason could be the differences in the atmospheric forcing with the two models forced by a different Evaporation-Precipitation rate. As stated earlier, further studies will be carried out to properly

assess the impact of the ECMWF forcing compared to the Met Office forcing.

The 1.5 km resolution model provides a better representation of dynamical features such as coastal currents, fronts and mesoscale eddies that can vary in size from only a few kilometres in shelf-seas to tens of kilometres, but a proper assessment is very difficult due to the high variability of these patterns and the very limited number of available observations.

There are some improvements in the tidal signal in AMM15 even if not so remarkable as with the salinity. One limitation is

the minimum depth set to 10m that prevents the model from properly taking into account the shallow bathymetry in the coastal areas. A wetting and drying implementation in under developments and could help could help to have a more realistic bathymetry, with improved tidal signal in very shallow waters, in a future version of AMM15.

The AMM15 ocean has been developed with coupled prediction in mind, the domain matching that of the Met Office atmospheric model UKV. Regional coupled model developments have been done and coupled ocean forecasting systems are

already planned.

The AMM15 system described in this paper, has been already tested in an ocean-wave coupled configuration (Lewis et al. 2018a) which is planned to become operational in 2020. We hope to add the biogeochemical components in a few years, but a precise plan is not yet available. Indeed, a preliminary version of AMM15 with coupled ocean-biogeochemistry is under development with first encouraging results but still far away from meeting the operational requirements. A coupled ocean

atmosphere version of this model has been already developed for research, Lewis et al. 2018b and studies will continue toward a fully coupled prediction system with ocean, atmosphere, land and wave model.

**Acknowledgements**

Funding support from the Copernicus Marine Environment Monitoring Service and the UK Ministry of Defence is gratefully acknowledged.

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
