# Peer review of "The impact of a new high-resolution ocean model on the Met Office North-West European Shelf forecasting system"

_Ocean Science, 2019_

## Referee Comment (RC1) · Anonymous Referee #1 · 22 Feb 2019

General comments

The paper "The impact of a new high-resolution ocean model on the Met Office North-West European Shelf forecasting system" presents in a really useful and interesting way the main components of the high resolution regional ocean forecasting system and the validation protocol and results. Main novelties and innovative works in this study concern the high resolution of this regional forecasting system including data assimilation of the main available observations. As mentioned by the authors, it seems difficult to exhibit really significant improvements link to the higher resolution especially because the validation protocol is based on standard comparison between model and

observations even if authors used specific high resolution observations based on gliders or HF radars. Nevertheless the study present an exhaustive comparison to available observations (assimilated or not) and validation diagnostics for most of the physical variables, these information are really useful for users of these operational forecast products and for developers of ocean forecasting system. I recommend the publication of this paper if the following minor revisions are taking into account in the final version.

1. Introduction

1. It could be useful to have a schematic view of the operational schedule of the system. The figure 2 with more information for example

2. Could you provide more precise information on the number of observations assimilated in the system thanks to the chosen assimilation cycles?

3. You mentioned the on going development of physic-biogeochemistry coupled system and the operational constrain. It's not the topic of the paper, but I suggest there is too much or not enough information for readers. Could you add few words about the time constrain and what kind of development is expected to reach the goal.

2. System Development

2.1 Core model Description

1. One specificity of the model configuration is the vertical coordinate system based on z*-$\sigma$. There is no justification in the description paragraph concerning the number of vertical levels which is the same than in the lower resolution system. Is there theoretical or experimental justification to reduce the rmax coefficient to 0.1 in this high resolution configuration and what is the expected impact (except the numerical stability)?

2. You impose a minimum of 10m depth on the bathymetry (this characteristic is also mentioned in the conclusion as a limitation), could you justify this choice, is only due to model stability?

3. How do you justify such difference (2 orders of magnitude) between the diffusion coefficient on tracer and advection?

2.1.1 Boundary and surface forcing

1. Could you add in the table 2 information concerning the difference of solar flux penetration in the two configurations and information on the tidal forcing at lateral boundaries

2.2 Assimilation method

Some information are missing in the description: 1. How is implemented the IAU method?

2. What is the SLA bias correction?

3. How do you use the 2 correlation length scale in the in the assimilation scheme? Do you perform 2 analysis?

4. In table 4, what are the differences between the 2 in situ data sources. How do you manage observation available in the two data bases?

5. In table 4, there is no information on the mean dynamic topography used to assimilate the SLA.

6. There is no information on methodology applied to assimilate the SLA in the model including tides.

2.3 Operational production

1. How is computed the QC error threshold for the observations?

2. You provide output fields on a standard vertical grid, how do you provide the information at the surface (0m) ? Is there a specific extrapolation to the surface?

3. Additional information concerning computing resources for this operational system could be useful (number of CPU, computer characteristics . . .)

4. More information could be added on figure 2 as for example, the observations, the atmospheric forcing, the restart and the assimilation and forecast sequence

4 Validation

4.1 Tides

1. M2 is the dominant tidal signal and probably the most important in an operational system for applications, user needs . . . One unexpected result increasing the resolution is perhaps the degradation of the mean M2 solution. It will be important in this section to discuss this point and highlight origin of this degradation.

4.2 Sea Surface Height

The section concerning SSH, as it is, is not really useful and could be removed. But as the SSH is assimilated in the system it's important to quantify impact of these observations. I suggest to add few diagnostics in comparison to SSH as for example :

Statistic/comparison with altimetry in open ocean where observation are assimilated. Along track comparison could be performed. It's important to understand in the paper why the SLA is assimilated in the system

Spatial power spectra to quantify spatial resolution of the system

Variability or eddy kinetic energy

4.3 Sea Surface Temperature

Temporal variability from seasonal cycle to high frequency is validated comparing model output to satellite observations and in situ time series. As expected there are few differences between the two models, main difference between the models being the horizontal resolution, even if the authors exhibit interesting higher frequency processes in the high resolution system. Even if it is not feasible with the observations why any spatial power spectra (or other diagnostics) has been performed to quantify

differences between the 2 models?

**4.4 Water Column**

On figure 10 larger bias and larger differences between AMM15 and AMM7 is located at 1000m depth. Is it linked to Mediterranean water? How do you explain this difference if the two configurations have the same constrains at the boundary and assimilates the same observations?

**4.4.2 Moorings in German Bight**

Few more information or hypothesis will be useful to explain some descriptions. - "The high frequency is better reproduced". Do you compute the correlation between the 2 time series? It's not so clear on figure 11

- "at the bottom AMM15 is more accurate". Why? Is it link to the bathymetry or link to vertical projection of increments?

- Table 8 : what is the depth of the bottom at each Buoy position?

- Figure 9 : why there is no model information in October? Add the correlation on the figure

**4.4.3 Glider transects**

Could you precise if the glider observations are assimilated or not in the system

**4.4.4 Mixed layer depth**

I suggest adding the mixed layer depth for AMM15 and AMM7 on figure 15 for example.

**4.5 Currents**

The comparison with HF radar observations is very useful and seems to be more relevant to compare high and low resolution model outputs. I suggest adding the statistics (mean, rms, correlation on amplitude and direction) which seems to be encouraging for the high resolution model as it is explain in the text but without the figures.

Figure 17 is nice to exhibit differences between the 2 models. It could be even better to add map with high resolution observations on the same area. Is there any SLA, SST or ocean color map that can be used to compare front and meso scale structures?

5 Conclusions and future developments

Something is missing in the conclusion, even if it is not obvious to validate and quantify improvement link to the higher resolution a discussion on expected improvements and link with user needs on this domain will be useful.

Typo, figures or format correction

1. Section Boundary and Surface Forcing should be 2.2 and then 2.3 Assimilation method, 2.4 operational system

2. Table 4 is cited before table 3.

3. Conclusion l 7 spatial/temporal
* * *

---

## Editor Comment (EC1) · Markus Meier (Editor) · 8 May 2019

Comments on the manuscript: *"The impact of a new high-resolution ocean model on the Met Office North-West European Shelf forecasting system". by M. Tonani et al.*

The manuscript presents a detailed description of a new MetOffice ocean forecasting system at 1.5Km. This new high-resolution AMM15 system, together with the previous existing one AMM7 (at 7Km resolution), complete the physical ocean model system used to produce the CMEMS North-West-Shelf ocean forecast and analysis product. A comprehensive validation of both systems is provided. To achieve this validation, a comparative assessment of both model systems has been performed, using trial runs over 2 years period.

As the authors mention in the manuscript, in some occasions it is not an easy task to demonstrate the significant improvement of higher resolution model performances. The difficulties to assess the differences between both model systems, the higher and lower resolution ones, are mainly related to the scarcity of adequate observational data sources. Nevertheless, the present paper aims to do it, and it presents a complete general validation work. Besides, it is shown some additionally examples of model validation with very specific (but geographically limited) observational data sources, such as gliders and HF radar sites.

Despite the general scientific interest of the manuscript may be enhanced, the proposed paper is of interest in the context of the present CMEMS OS special issue. The complete description and exhaustive assessment of the new High-Resolution model system with respect to the previous existing one, is of interest for future, scientific and non-scientific, CMEMS NWS end-users, using products derived from the model systems here presented. Therefore, I do recommend publication of the manuscript after revision of some points.

For instance, I would ask the authors to justify in the revised manuscript some of the decisions taken to build the new 1.5Km model system set-up. The authors should address in more detail some of the choice made related to:

1) the model configuration (i.e. why the authors keep in the high-resolution system the same tidal forcing (using the same 12 harmonics) then in the lower resolution system, why they use the same vertical grid distribution)
2) the data assimilation scheme used in the AMM15 system (i.e. why in a shelf model system, as the AMM15 is, it is assimilated SLA only outside the shelf; and how do the authors face the challenge of assimilating altimetric observations in high tidal environments).

Providing more info on these points, of interest for ocean shelf modelers, the authors certainly will enhance the scientific interest of the paper for the ocean data assimilation and modelling community.

Apart of the points to be address in more detail by the authors, there are also some suggestions of text changes to improve the paper readability. For instance, I would recommend moving any "pure" model-model comparisons, that is with no observational data source used as reference, from the Section 4, dedicated to validation results. Thus, the final comparisons shown on the surface currents patterns may be moved to another earlier section. I find these results very illustrative and give a good measure of the differences that we can expect from the new increased resolution model system (so, they

should be included in the manuscript to show the different model performance achieved), but they do not provide any model validation (so, this text should be out of Section 4).

As it is said before, I recommend publication of the manuscript after revision of the following points listed below.

**Abstract:**

P1.15 "… (AMM7) that has been used for many years". Please, specify the context (CMEMS?, before Copernicus?)

P1.18 "Trial experiments run with the low and high resolution systems in their operational configuration". Please, specify if this operational configuration includes Data Assimilation, or means just forecast runs.

**Section 1. Introduction:**

P1.35 In this paragraph, the authors mention human activities (industrial, farming, fishing) with climate change as source of impacts in the quality of water environments. All of them have certainly an impact, but I would suggest re-drafting the sentence, separating the impacts from climate change and the human-related activities, since they are at different levels.

P2.1 Include some reference to sustain the paragraph.

P2.13 The (CMEMS?) operational forecasting for the North-West European Shelf (NWS). Other applications?

P2.15 To describe the geographical domain, the Figure 1 is referred. However, when a reader goes to this Figure, sees 2 different model domains: the AMM15 & AMM7, not mentioned yet and with not defined acronyms. It is a bit confusing for the reader at a first reading. The authors should improve this point: 1) moving after in the text the citation of this Figure 1, or 2) improving the figure caption to give more information on the features shown.

P2.24 From this line up to the end of this Introduction Section, the authors mention different components of the CMEMS NWS system (i.e: ocean physical model, data assimilation system, together with the biogeochemical model coupled into it). Also, there is a mention to a wave model system, and to an ocean-wave-atmospheric coupled system. In order to enhance the understanding of the systems and its multiple connections with other applications, here outlined, the authors should include a figure showing a schematic view of the CMEMS NWS operational forecast system, here described. This extra figure suggested may be included as part of the present Figure 2. This way, the number of figures is not increased and the present Figure 2, what currently provides certainly very few information, is enhanced.

I would also miss in this part of the manuscript some reference to the CMEMS operational products generated through the model systems here described (with citation to their documentation).

P2.30 With respect to the latest system mentioned in this paragraph: the ocean-wave-atmosphere coupled system, is this currently an operational one? or is it in a pre-operational phase? Or only for research purposes?

P2.37 Please, include some quantitative numbers or estimations to support the adjective "prohibitive".

**Section 2. System Description**

P3.10 Use CMEMS instead Copernicus.

P3.19 The new AMM15 system uses the same vertical grid resolution than the AMM7 one. Why it was not considered to increase the vertical resolution consistently with the horizontal one? Is the present vertical resolution with 51 levels enough? Have the authors performed any sensitivity test to evaluate the impact of an enhancement of vertical resolution? Or the decision to keep the vertical resolution unchanged is more a matter of computational resource availability? Any comment on this point?

P4 No reference in the text to Table 1?

P5.9 Are 12 tidal harmonic constituents enough to rightly reproduce the tides in a region such as the one covered by the high resolution AMM15 model, that is marked by shelf shallow waters with very high tidal environments? Can the authors justify why the same 12 harmonics are used in both systems? Since the objective is to model the region at a very high resolution, it would not be worthy to count with an improved higher resolution tidal forcing (the original TPX harmonic are at a 1/12° resolution). Furthermore, please, include in the manuscript the list of the 12 harmonics used (this list of harmonics can be provided directly in the text, or in Table 2).

P5.18 The authors mentioned that ECMWF IFS data is used as forcing in the AMM15 system, whereas the AMM7 uses the MetUM forcing. The move to the ECMWF forcing is justified as a requirement of the CMEMS service. However, the IFS data have lower resolution than the MetUM (around 14 Km in the former, instead of the 10 Km of the later). Apart of this "service" reason, can the authors comment on the impacts that move from a higher resolution forcing to a lower one has in the ocean model solution?
Furthermore, later the authors mention that using IFS there is a lost in terms of analysis frequency availability (from 3h to 6h). Can the authors provide some quantification of the impact related to the change in the forcing? It is certainly not very intuitive for a reader to understand how when a new higher resolution model system is being set up, it is decided to use a lower resolution atmospheric forcing. Can the authors explain any positive impact of the change in the atmospheric forcing to support the decission?

P7.11 It is stated that there is SLA assimilation both in MM7 and MM15 systems, and in both cases, for regions with bathymetric depths > 700m. In the case of the MM7 configuration this option can make sense, since extended deep water areas are covered. However, on the contrary in the case of the AMM15 shelf model system, this set-up option seems to result in a SLA data assimilation limited to a very narrow area (and very close to the open boundaries!). Can the authors explain in more detail the impact of the SLA data assimilation approach performed on the AMM15 shelf system? Can the authors provide a measure of the benefit of assimilating SLA data assimilation on such a limited (and so close to the boundaries) area? The authors should explain better the potential gain of using such limited SLA data assimilation with respect to a free non-assimilative approach.

P7.13 Table 4 cited before Table 3. Please, try to respect the order in Figure and Table citation.

Table 4. In the column of Data source: 1st arrow: "CMEMS –INS-TAC" may be substituted by "CMEMS-INS-TAC Product:" The same for "GTS" ("GTS Product:"?); 2nd arrow: "CMEMS-SL-TAC Product:"
3rd arrow: "Product from the Group for High Resolution Sea Surface Temperature (GHRSST):"

P9.15 What is it done with the info on the profile quality check performed? Any communication established with the observational data producers? (a kind of blacklisting?).

P9.19 Do the authors foresee any problem in using OBCs from different model data sources? Are they consistent? Can be a source of problems due to volume conservations issues?

P9.37. The production process takes approximately 4 hours. How many CPUs are used during the process? Can the authors include here a computational cost estimation?

Figure 2: Include here info on the ECMWF IFS forcing (analysis/forecast) used. Complete this Figure, as suggested in previous comment, showing a schematic view of the CMEMS NWS operational forecast system described.

**Section 3. Trial experiments**

Figure 3. Number of observations used for assimilation. The panel on the SLA show effectively the satellite SLA observations available. However, this panel can mislead the reader, since the data assimilation is applied only on areas with depths > 700m. I suggest the authors will identify in the plot the area where SLA is effectively assimilated in AMM15 system.

The authors should consider the possibility to include in this section the analysis of the differences between the dynamical patterns modelized by the 2 different model

systems, currently included in the Validation Result section. This point is suggested below.

**Section 4. Validation of the experiments**

Figure 4. specify also here the locations where observations from coastal tide gauges are available for the tidal validation.

2 figures are dedicated to display location of observational data sources used in the paper. The Figure 3 shows those observations used in the data assimilation. On the other hand, the Figure 4 displays other observational data sources used in the validation process. Where are the coastal tide gauges? I guess they are not assimilated, however they are not depicted neither in Figure 4.

Furthermore, the reader founds later in Table 6 (where results from the validation of different variables are shown) results for the M2 tidal harmonic and there it is said that validation is done for the full domain. However, no info on the location of the tide gauges used is provided up to that moment. Later, already in Section 4.1, in the Figure 5 there is a map of model-obs differences in M2 amplitude and phase. Please, clarify a bit the geographical information on the tide gauge locations.

P14.11 Tide gauges observations from BODC. "The number of tides gauges taken into consideration for AMM15 and AMM7 is the same, therefore the coastal buoys"; are the authors here referring to tide gauges? Or to buoys? Can the authors provide more details about the tidal observations offshore, where do they come from? (From platforms?, pressure sensors?). More explanation about the tidal measurements from the BODC it may help the reader.

P15.13 Suggestion to ease readability: in one of the maps, for instance in Figure 1, the authors should detail all the geographical names mentioned in the text (i.e. German Bight). This reference to geographical features will ease the reading of the paper to those potential readers not familiar with the regional geography.

P16.5 The authors shown in the paper (section 4.1.2) some results of the model validation with a HF Radar system. The results are only for 1 month (March 2017). If the authors have 2 years model runs, why do they perform/show a so short temporal coverage of the model- HF Radar validation? Due to observation availability? Please, explain reasons in the manuscript.

Figure 6 shows the results of the model-HFRadar validation. In this figure, it is shown some statistics fields (RMSD, Bias, Veering) limited to the HF Radar spatial coverage. However, the reader have no information about the number of observational data that support these statistics. Do the HF-Radar system provide exactly the same number of observations everywhere? If yes, please detail what gap filling methodology is being applied. If not, please, show the % of HF-Radar data availability.
I guess the 3 names referred in this figure 6 corresponds to the HF Radar sites. Please, detail in the Figure Caption.

P18.13 The in-situ measurements are from buoys and ships of opportunity. Please, detail if "buoys" means fixed moorings, surface drifters or ARGO profilers.

P19.4 A Butterworth filter. Please, explain in more detail or add a reference.

P19.20 AMM7 and AMM15 models provide very similar values of SST, probably due to the data assimilation of SST that brings models close to the observations. Can the authors include in the paper any SST timeseries analysis as the one here shown for the 3 proposed sites, but in a station, whose SST observational data would not be assimilated? See some independent validation would certainly be of interest for readers and potential users of the model products.

Figure 9. It is quite remarkable the overestimation of the 12-h energy peak in AMM15. Any relation with the harmonic bias in M2? It is also interesting the notorious AAM15 peak around 6-h. Can the authors comment on it? Any explanation? May it be linked to the meteorological forcing? (different in both model systems).

P23.1 E-Hype. What is E-Hype? No mention to this name in the section where forcing are described. Please, introduce complete name of the source or reference.

P24.24 Please, check the date: 23rd March or 23rd May (as referred in the Figure 12 caption; in this Fig 13 caption, correct typo: 23rt).

In Section 4.4.4 it is discussed about model differences in MLD and it is referred to the Figure 15, where only the MLD computed from the glider is depicted. Why the MLD computed from the models are not shown in the panels Glider-MM15 & Glider-MM7 together with the one derived from the glider data? Include the MLDs from both models in the plot can enhance the analysis in this section dedicated to MLD.

Section 4.5 is devoted to show some results from currents compared with HF-Radar data. As in the previous case for the tides, only a month of data (March 2017) is shown. Please, justify why a so short temporal coverage for the validation.
Figure 16 shows monthly values of the HF-Radar and from the 2 models, interpolated to the observational field. However, no information on how many observations support the resulting monthly value is provided. Please, include the % of data availability for the month shown. It will be also useful to have some information on the validation of the HF-Radar measurement, as well as on the gap filling methodology used (if someone is used). The explanation/discussion of the comparative results is quite poor. Please, provide some more description of the features depicted. For instance, it will be interesting that the authors describe the high currents feature existing in front of Wang and Busum stations, reproduced by the AMM15, but not for the AMM7 model. Likewise, any explanation or comment about possible border effects in the HF-Radar field shown would also be pertinent. Can the authors ensure that all the high currents depicted at the border of the HF Radar coverage are reliable? Please, include some info in the text (a reference would also help) on the existing validation of the HF Radar data used and about the possibility of border effects in the observational data used.

The analysis of the AMM7 & AMM15 model currents provided from P 28.14 till the end of the Section 4.5 (including reference to Figure 17) is not referred to any model validation. It is not used any observational data source used as reference. Therefore, I would suggest taking this analysis out from this Validation section.

I found the analysis interesting, and it illustrates quite well the dynamical differences existing between both model solutions.

If the authors want to keep this analysis in the manuscript, I would suggest moving this part of the text and the figure to the end of the Section 3 (where Trial experiments are described). This analysis of the dynamical patterns obtained gives a good idea of how different the 2 model solutions are and it may give a good introduction to the reader to the validation results that come later in Section 4.

**Section 5. Conclusions and future developments.**

P30.7 typo: temporal

Please, include in the conclusion section some reference to the Data Assimilation performed in the AMM15 system, with mention to potential future plans to enhance the assimilation process (and very specially for SLA on the shelf).

P31.18 The AMM15 ocean (system?).

---

## Author Comment (AC1) · 19 Jun 2019

Manuscript title: *"The impact of a new high-resolution ocean model on the Met Office North-West European Shelf forecasting system". by M. Tonani et al.*

**Bold: referee's comment**
Not bold: author's answer

The referee's comments are copied in this document for ease of reading.

General comment:
**The manuscript presents a detailed description of a new MetOffice ocean forecasting system at 1.5Km. This new high-resolution AMM15 system, together with the previous existing one AMM7 (at 7Km resolution), complete the physical ocean model system used to produce the CMEMS North-West-Shelf ocean forecast and analysis product. A comprehensive validation of both systems is provided. To achieve this validation, a comparative assessment of both model systems has been performed, using trial runs over 2 years period.**
**As the authors mention in the manuscript, in some occasions it is not an easy task to demonstrate the significant improvement of higher resolution model performances. The difficulties to assess the differences between both model systems, the higher and lower resolution ones, are mainly related to the scarcity of adequate observational data sources. Nevertheless, the present paper aims to do it, and it presents a complete general validation work. Besides, it is shown some additionally examples of model validation with very specific (but geographically limited) observational data sources, such as gliders and HF radar sites.**

**Despite the general scientific interest of the manuscript may be enhanced, the proposed paper is of interest in the context of the present CMEMS OS special issue. The complete description and exhaustive assessment of the new High-Resolution model system with respect to the previous existing one, is of interest for future, scientific and non-scientific, CMEMS NWS end-users, using products derived from the model systems here presented. Therefore, I do recommend publication of the manuscript after revision of some points.**

**For instance, I would ask the authors to justify in the revised manuscript some of the decisions taken to build the new 1.5Km model system set-up.**
**The authors should address in more detail some of the choice made related to:**

**1. The model configuration (i.e.. why the authors keep in the high-resolution system the same tidal forcing (using the same 12 harmonics) then in the lower resolution system. Why they use the same vertical grid distribution)**

The major aim of this model configuration is to resolve the Rossby Radius on the shelf, therefore the focus was on increasing the resolution from 7 to 1.5 km. All the technical details of the implementation and the validation of the model, without data assimilation, are presented in Graham et al., 2018. That is the precursor work to this paper. The tidal forcing is going to be improved, in the next release of the AMM15 model that will include also the wetting and drying. Experiments are ongoing, using FES2014 and more tidal constituents. The number of vertical levels is the same because the focus of this model is on the shelf (depth < 200m), where 51 z-sigma levels are enough for proving a very high vertical resolution. The resolution is of the order of 20cm the shallower part of the model domain, where the minimum depth is 10. More levels will increase the model vertical resolution in the deepest part of the domain, not on shelf (Siddorn et al., 20016). Another possible approach is using

vertically adaptive vertical coordinates so that you focus resolution on the thermocline. This is done in other models but not here and will be considered in the future configurations.

It's important to have a step by step incremental improvements protocol for updating an operational system, rather than changing many things all at once. This is the very first implementation of AMM15 in operations. This system or part its components will be improved on a yearly base in the future releases.

We have provided more technical information on this topic answering the specific questions here below.

2. **The data assimilation scheme used in the AMM15 system (i.e. why in a shelf model system, as the AMM15 is, it is assimilated SLA only outside the shelf; and how do the authors face the challenge of assimilating altimetric observations in high tidal environments)**

**Providing more info on these points, of interest for ocean shelf modelers, the authors certainly will enhance the scientific interest of the paper for the ocean data assimilation and modelling community.**

This first implementation of assimilation in the high resolution AMM15 followed the same scheme as used in the AMM7. Although the deep water areas of AMM15 are more limited than AMM7, and so the benefit of assimilating SLA may be small, this is a milestone on the way to assimilating SLA throughout the domain.
However, the assimilation of SLA observations in the deeper water still allows us to constrain the temperature and salinity in those regions, which then provides a better boundary condition for the shallow regions. As discussed in King et al. 2018, the assimilation of altimeter observations and T/S profiles is complementary, and in regions such as the NWS where profile observations are relatively limited, altimeter observations provide a valuable additional constrain on the density structure of the deep water regions.

Suggestions of text changes to improve the paper readability
**I would recommend to moving any "pure" model-model comparisons, that is with no observational data source used as reference, from the Section 4, dedicated to validation results. Thus, the final comparisons shown on the surface currents patterns may be moved to another earlier section. I find these results very illustrative and give a good measure of the differences that we can expect from the new increased resolution model system (so, they should be included in the manuscript to show the different model performance achieved), but they do not provide any model validation (so, this text should be out of Section 4)**

We thank the reviewer for this suggestion. We have modified the manuscript moving the "pure" model-model comparison in section 3:"*3.1 System performance: AMM5 vs AMM7*".
The sub-section on the "Currents" has been rename as "Currents in the German Bight" and moved up after the "Tidal flow" sub-section, because both based on the same HF radar observations.

The manuscript is now organized as follow:
*Abstract*
1. *Introduction*
2. *System description*
    2.1 *Core model description*

**As it is said before, I recommend publication of the manuscript after revision of the following points listed below.**

Abstract

**P1.15 "… (AMM7) that has been used for many years". Please, specify the context (CMEMS?, before Copernicus?)**

We added this information in the text.

OLD:

"*The latest configuration to be put in operations, an eddy resolving model at 1.5 km (AMM15), replaces the 7km model (AMM7) that has been for a number of years.*"

NEW:

"*The latest configuration to be put in operations, an eddy resolving model at 1.5 km (AMM15), replaces the 7km model (AMM7) that has been used for eight years to deliver forecast products to the Copernicus Marine Service and its precursor projects.*"

**P1.18 "Trial experiments run with the low and high resolution systems in their operational configuration". Please, specify if this operational configuration includes Data Assimilation, or means just forecast runs.**

The sentence has been reworded to specify that the trial experiments are done with data assimilation:

"*Validation of the model with data assimilation is based on the results of two years (2016-2017) trial experiments run with the low ……*"

1.Introduction

**P1.35 In this paragraph, the authors mention human activities (industrial, farming, fishing) with climate change as source of impacts in the quality of water environments. All of them have certainly an impact, but I would suggest re-drafting the sentence, separating the impacts from climate change and the human-related activities, since they are at different levels.**

We have re-worded that sentence as follow:

*"The increasing focus on understanding the marine environment in support of sustaining healthy and biologically diverse seas is also a considerable driver in these waters, where human activities like heavy industrial and farming activity, as well as fishing together with climate change effects, may have significant impacts on the quality of the marine environment."*

**P2.1 Include some reference to sustain the paragraph.**

We added the following references :

She, J., Allen, I., Buch, E., Crise, A., Johannessen, J. A., Le Traon, P.-Y., Lips, U., Nolan, G., Pinardi, N., Reissman, J.H., Siddorn, J., Stanev, E., Wehde, H.: Developing European operational oceanography for Blue Growth, climate change adaptation and mitigation, and ecosystem-based management, Ocean Science, 12(4) 953-976 https://doi.org/10.5194/os-12-953-2016, 2016.

Siddorn, J.R, Good, S. A., Harris, C. M., Lewis, H. W., Maksymczuk, J., Martin, M. J., Saulter, A.: Research priorities in support of ocean monitoring and forecasting at the Met Office, Ocean Science, 12(1), https://doi.org/10.5194/os-12-217-2016, 2016.

**P2.13 The (CMEMS?) operational forecasting for the North-West European Shelf (NWS). Other applications?**

This operational configuration has been implemented for CMEMS. Several CMEMS users are using the product for a wide range of applications or for developing downstream products.

**P2.15 To describe the geographical domain, the Figure 1 is referred. However, when a reader goes to this Figure, sees 2 different model domains: the AMM15 & AMM7, not mentioned yet and with not defined acronyms.  It is a bit confusing for the reader at a first reading. The authors should improve this point: 1) moving after in the text the citation of this Figure 1, or 2) improving the figure caption to give more information on the features shown.**

Thanks for the suggestion, we opted for improving the caption of figure1 :

*Figure 1: EMODnet bathymetry, in meters (logarithmic scale), showing the NWS high resolution, AMM15, model domain. The red line defines the NWS low resolution, AMM7, model domain. The yellow dotted box is the domain covered by the AMM15 products delivered on a regular grid to the Copernicus users. Figure modified from Graham et al. (2018). The bathymetry colour range has logarithmic scale.*

**P2.24 From this line up to the end of this Introduction Section, the authors mention different components of the CMEMS NWS system (i.e: ocean physical model, data assimilation system, together with the biogeochemical model coupled into it). Also, there is a mention to a wave model system, and to an ocean-wave-atmospheric coupled system. In order to enhance the understanding of the systems and its multiple connections with other applications, here outlined, the authors should include a figure showing a schematic view of the CMEMS NWS operational forecast system, here described. This extra figure suggested may be included as part of the present Figure 2. This way, the number of figures is not increased and the present Figure 2, what currently provides certainly very few information, is enhanced.**
**I would also miss in this part of the manuscript some reference to the CMEMS operational products generated through the model systems here described (with citation to their documentation).**

Thanks for this suggestion. We had added this information in figure 2:

[Figure]

We have added the list of the CMEMS products at the end of the introduction:

"*The7km products (AMM7) delivered though Copernicus are:*

- *NORTHWESTSHELF_ANALYSIS_FORECAST_PHY_004_001_b*
  (*http://marine.copernicus.eu/documents/PUM/CMEMS-NWS-PUM-004-001.pdf*);

- *NORTHWESTSHELF_ANALYSIS_FORECAST_BIO_004_002_b*
  (*http://marine.copernicus.eu/documents/PUM/CMEMS-NWS-PUM-004-002.pdf*).

*The1.5km products (AMM15) delivered though Copernicus are:*

- *NORTHWESTSHELF_ANALYSIS_FORECAST_PHY_004_013*
  (*http://marine.copernicus.eu/documents/PUM/CMEMS-NWS-PUM-004-013.pdf*);

- *NORTHWESTSHELF_ANALYSIS_FORECAST_WAV_004_014*
  (*http://marine.copernicus.eu/documents/PUM/CMEMS-NWS-PUM-004-014.pdf*).

*This study is focused on the product NORTHWESTSHELF_ANALYSIS_FORECAST_PHY_004_013 and its inter-comparison with NORTHWESTSHELF_ANALYSIS_FORECAST_PHY_004_001b.*"

**P2.30 With respect to the latest system mentioned in this paragraph: the oceanwave-atmosphere coupled system, is this currently an operational one? or is it in a pre-operational phase? Or only for research purposes?**

This is system is just for research purposes at present, as written in the text "*coupled ocean-wave-atmosphere research system*".

**P2.37 Please, include some quantitative numbers or estimations to support the adjective "prohibitive".**

We added this sentence to the manuscript:
*", because the production time exceeds the 24-hr"*

2.System Description

**P3.10 Use CMEMS instead Copernicus.**
Done

**P3.19 The new AMM15 system uses the same vertical grid resolution than the AMM7 one. Why it was not considered to increase the vertical resolution consistently with the horizontal one? Is the present vertical resolution with 51 levels enough? Have the authors performed any sensitivity test to evaluate the impact of an enhancement of vertical resolution? Or the decision to keep the vertical resolution unchanged is more a matter of computational resource availability? Any comment on this point?**

The increased horizontal resolution moving from AMM7 to AMM15 allows a step change in the ability to represent small-scale processes, but there remains work to be done to address biases in the vertical representation in the shelf seas (detailed in Graham et al. 2018a). These are influenced by many factors including the vertical mixing scheme, advection, light attenuation scheme, and wave-mixing parameterizations. Simply increasing the vertical resolution was not expected to lead to any improvements without first addressing these physical mechanisms (see also answer to the first general question).

**P4 No reference in the text to Table 1?**
Thanks, we added the reference to Table1 in the manuscript.

**P5.9 Are 12 tidal harmonic constituents enough to rightly reproduce the tides in a region such as the one covered by the high resolution AMM15 model, that is marked by shelf shallow waters with very high tidal environments? Can the authors justify why the same 12 harmonics are used in both systems? Since the objective is to model the region at a very high resolution, it would not be worthy to count with an improved higher resolution tidal forcing (the original TPX harmonic are at a 1/12º resolution). Furthermore, please, include in the manuscript the list of the 12 harmonics used (this list of harmonics can be provided directly in the text, or in Table 2).**

We thank the author for this comment. There is a typo in the manuscript, the tidal constituents are 11, not 12 (We have corrected the manuscript).
The first implementation of AMM15 as been set up as much as possible as AMM7 for understanding the impact of the increased resolution. The number of tidal constituents has not been increased.

AMM15 has less tidal constituents than AMM7 due to the different model used (see the updated Table 2 here below).

It is worth to note that the tidal boundaries in AMM are in the deep (off shelf) region for the most part (excepting short stretches where they cross the continental shelf). The higher modes are less important in the deep. They are significant only in the shallows where a large component of them are going to be locally generated by the interaction of the primary constituents with the bed and coastline rather than remotely forced at deep water boundaries.

Yes, the referee is correct, it is important to improve the tidal forcing of AMM15, in terms of tidal constituents and atlases. Research activities are ongoing to validate the impact of using a different model, FES2014, with many more tidal constituents.

The updated version of Table 2 is:

| Forcing | AMM7 | AMM15 |
|---|---|---|
| **Surface forcing** | Met Office Global Unified Model (MetUM) Atmospheric model NWP analysis and forecast fields, calculated in the MetUM using COARE4 bulk formulae (Fairall et al. 2003). | ECMWF Integrated Forecasting System (IFS)-Atmospheric Model High Resolution (HRES) operational NWP forecast fields using CORE bulk formulae (Large and Yeager 2009) |
| **Surface forcing resolution** | Horizontal grid: ~10 km (2560 x 1920 grid points) Frequency: 3 hourly mean fluxes of long and short wave radiation, moisture, 3 hourly mean air surface temperature but hourly 10m winds and surface pressure | Horizontal grid: ~14 km (0.125°x0.125°). Frequency: 3 hourly instantaneous 2m dew point temperature, surface pressure, mean sea level pressure, and 2m air temperature. 3 hourly accumulated surface thermal and solar radiation, total precipitation, and total snow fall. |
| **River run-off** | Daily climatology of gauge data averaged for 1950–2005. Climatology of daily discharge data for 279 rivers from the Global River Discharge Data Base (Vörösmarty et al., 2000) and from data prepared by the Centre for Ecology and Hydrology as used by (Young and Holt, 2007). | Daily climatology of gauge data averaged for 1980–2014. UK data were processed from raw data provided by the Environment Agency, the Scottish Environment Protection Agency, the Rivers Agency (Norther Ireland), and the National River Flow Archive (personal communication by Sonja M. van Leeuwen, CEFAS, 2016). For major rivers that were missing from this data set (e.g. along the French and Norwegian coast), data have been provided by the same climatology used by AMM7 (Vörösmarty et al., 2000 and Young and Holt, 2007). |
| **Tidal constituents** | M2, S2, N2, K2, K1, O1, P1, Q1, M4, MS4, L2, T2, S1, 2N2, MU2, NU2 (15) from a tidal model of the North-East Atlantic (Flather, 1981). | M2, S2, N2, K2, K1, O1, P1, Q1, M4, MS4, MN4 (11) from Topex Poseidon cross-over solution (Egbert and Erofeeva, 2002; TPX07.2, Atlantic Ocean 2011-ATLAS). |
| **Lateral boundaries** | Met Office FOAM North Atlantic (1/12°; 6 hourly fields) and CMEMS Baltic Sea (2km, 1 hourly fields). ||
| | AMM7 and AMM15 have Atlantic and Baltic boundaries in a different geographical location. ||

**P5.18 The authors mentioned that ECMWF IFS data is used as forcing in the AMM15 system, whereas the AMM7 uses the MetUM forcing. The move to the ECMWF forcing is justified as a requirement of the CMEMS service. However, the IFS data have lower resolution than the MetUM (around 14 Km in the former, instead of the 10 Km of the later). Apart of this "service" reason, can the authors comment on the impacts that move from a higher resolution forcing to a lower one has in the ocean model solution?  Furthermore, later the authors mention that using IFS there is a lost in terms of analysis frequency availability (from 3h to 6h). Can the authors provide some quantification of the impact related to the change in the forcing? It is certainly not very intuitive for a reader to understand how when a new higher resolution model system is being set up, it is decided to use a lower resolution atmospheric forcing. Can the authors explain any positive impact of the change in the atmospheric forcing to support the decision?**

Yes, we agree with the referee that we need to run impact studies to assess the impact of the different resolution in the atmospheric forcing. These studies were not part of the pre-operational implementation, therefore are not discussed in this paper. These experiments are carried out in the frame of a UK project. As soon as available these results will be shared with CMEMS. If the impact of the coarser spatial resolution is significant we could switch to the Met UM forcing in a future evolution of the operational system.
The ECMWF-IFS analyses are not used, we use only 3-hr forecast fields. We are planning to switch to hourly ECMWF-IFS products by the end of 2020.

**P7.11 It is stated that there is SLA assimilation both in MM7 and MM15 systems, and in both cases, for regions with bathymetric depths > 700m. In the case of the MM7 configuration this option can make sense, since extended deep water areas are covered.  However, on the contrary in the case of the AMM15 shelf model system, this set-up option seems to result in a SLA data assimilation limited to a very narrow area (and very close to the open boundaries!). Can the authors explain in more detail the impact of the SLA data assimilation approach performed on the AMM15 shelf system?  Can the authors provide a measure of the benefit of assimilating SLA data assimilation on such a limited (and so close to the boundaries) area? The authors should explain better the potential gain of using such limited SLA data assimilation with respect to a free non-assimilative approach.**

This first implementation of assimilation in the high resolution AMM15 followed the same scheme as used in the AMM7. Although the deep water areas of AMM15 are more limited than AMM7, and so the benefit of assimilating SLA may be small, this is a milestone on the way to assimilating SLA throughout the domain.
However, the assimilation of SLA observations in the deeper water still allows us to constrain the temperature and salinity in those regions, which then provides a better boundary condition for the shallow regions. As discussed in King et al. 2018, the assimilation of altimeter observations and T/S profiles is complementary, and in regions such as the NWS where profile observations are relatively limited, altimeter observations provide a valuable additional constrain on the density structure of the deep water regions. Without additional experiments, the contribution of altimeter observations is difficult to quantify.

**P7.13 Table 4 cited before Table 3. Please, try to respect the order in Figure and Table citation.**

Thanks, we swapped table 3 and table 4 and the corresponding cross-references.

**Table 4. In the column of Data source: 1st arrow: "CMEMS –INS-TAC" may be substituted by "CMEMS-INS-TAC Product:" The same for "GTS" ("GTS Product:"?); 2nd arrow: "CMEMS-SL-TAC Product:" 3rd arrow: "Product from the Group for High Resolution Sea Surface Temperature (GHRSST):"**

Thanks for this comment. We modified changed Table 4 taking into consideration this comment. The column "Data source" is now "Data source/Products":

| Type | Fields | Platforms/Satellite | Data source/Product |
|------|--------|---------------------|---------------------|
| IN SITU | SST

Temperature and salinity profiles | • Ships
• Drifters
• Fixed moored arrays
• Gliders
• XBTs
• CTDs
• ARGO
• Ferry boxes
• Recopesca buoys*
• Thermosalinograph | GTS;
http://www.wmo.int/pages/prog/www/TEM/GTS

http://marine.copernicus.eu/
INSITU_GLO_NRT_OBSERVATIONS_01
3_030 |
| SATELLITE | SLA Along Track*
*along with the corrections necessary for the use with a wind and pressure forced, tidal coastal model. SLA are assimilated only in deep regions (> 700m). | • Cryosat-2
• Altika
• Jason 3
• Sentinel 3a | http://marine.copernicus.eu/
SEALEVEL_EUR_PHY_ASSIM_L3_NRT
_OBSERVATIONS_008_043 |
| | SST L2p/L3c | • NOAA-AVHRR
• MetOp-AVHRR
• SEVIRI
• VIIRSG
• AMSR2 | Group for High-Resolution Sea Surface Temperature (GHRSST)
www.ghrsst.org |

**P9.15 What is it done with the info on the profile quality check performed? Any communication established with the observational data producers? (a kind of blacklisting?).**

We store all the information on the quality check in a set of files called "feedback files" (each for each type of observations: sub-surface profiles of temperature and salinity; SLA; SST). We are working with the CMEMS Product Quality Cross-Cutting working group to identify a CMEMS standard for conveying this information to the data producers.

**P9.19 Do the authors foresee any problem in using OBCs from different model data sources? Are they consistent? Can be a source of problems due to volume conservations issues?**

The two models providing the boundaries are not consistent and this is the reason why we don't force the Baltic Boundary with SSH (The Baltic model is not constrained by data assimilation while the North Atlantic is). The Atlantic model providing the boundaries and AMM are both constrained by the assimilation of the same SLA data, this should avoid major discrepancies at the Atlantic boundary. The forcing at the boundaries is an active research topic for our model and hopefully in the future we can improve the parametrisation we are using now even if the problem is more

worrying when producing reanalysis or climate simulations spanning over a much longer number of years than a short-term forecast system.

**P9.37. The production process takes approximately 4 hours. How many CPUs are used during the process? Can the authors include here a computational cost estimation?**

These operational systems are running on the Met Office HPC – Cray XC40 super computer. The following table describes the number of nodes and processors used by each component:

| System | Component | # of nodes | # of processors |
|--------|-----------|------------|-----------------|
| AMM7 | NEMO | 8 | 256 |
| | XIOS | -- | -- |
| | NEMOVAR | 2 | 64 |
| AMM15 | NEMO | 48 | 1536 |
| | XIOS | 8 | 256 |
| | NEMOVAR | 48 | 1536 |

XIOS is for the NEMO I/O. The small size of AMM7 model grid doesn't require dedicated nodes for this task.

We added this information to the manuscript.

**Figure 2: Include here info on the ECMWF IFS forcing (analysis/forecast) used. Complete this Figure, as suggested in previous comment, showing a schematic view of the CMEMS NWS operational forecast system described.**

Yes, the figure has been updated, please see comment P2.24
We use only the forecast fields from ECMWF-IFS, due to the coarse time resolution of the analysis (6-hr), as specified in table 2 and at P6.2 of the manuscript: "*The IFS analysis is available only at a low temporal resolution (6 hours) therefore the decision was made to force the system using forecast fields only (3 hourly), from the 00:00 UTC forecast base time*".

3.Trial Experiments
**Figure 3. Number of observations used for assimilation. The panel on the SLA show effectively the satellite SLA observations available. However, this panel can mislead the reader, since the data assimilation is applied only on areas with depths > 700m. I suggest the authors will identify in the plot the area where SLA is effectively assimilated in AMM15 system.**

Thanks for the comment, the new version of the Figure 3 shows now only the assimilated SLA obs:

[Figure]

The authors should consider the possibility to include in this section the analysis of the differences between the dynamical patterns modelized by the 2 different model systems, currently included in the Validation Result section. This point is suggested below.

Yes, thanks for the comment. We have taken this suggestion into consideration as described in detail at the beginning in the general comments.

4.Validation of the experiments

**Figure 4. specify also here the locations where observations from coastal tide gauges are available for the tidal validation.**

Thanks, Figure 4 has been updated with the location of all the tide gauges used for the tidal validation (yellow dots).

[Figure]

**2 figures are dedicated to display location of observational data sources used in the paper. The Figure 3 shows those observations used in the data assimilation. On the other hand, the Figure 4**

displays other observational data sources used in the validation process. Where are the coastal tide gauges? I guess they are not assimilated, however they are not depicted neither in Figure 4. Furthermore, the reader founds later in Table 6 (where results from the validation of different variables are shown) results for the M2 tidal harmonic and there it is said that validation is done for the full domain. However, no info on the location of the tide gauges used is provided up to that moment. Later, already in Section 4.1, in the Figure 5 there is a map of model-obs differences in M2 amplitude and phase. Please, clarify a bit the geographical information on the tide gauge locations.

Thanks for pointing out this inconsistency. Yes, it's correct, the tide gauges data are not assimilated. We have updated Figure 4 adding the tide gauge location (see the figure in the comment above).

**P14.11 Tide gauges observations from BODC. "The number of tides gauges taken into consideration for AMM15 and AMM7 is the same, therefore the coastal buoys"; are the authors here referring to tide gauges? Or to buoys? Can the authors provide more details about the tidal observations offshore, where do they come from? (From platforms?, pressure sensors?). More explanation about the tidal measurements from the BODC it may help the reader.**

Thanks for the comment, it's not appropriate calling "buoys" the tide gauges data. We have corrected the manuscript as follow:

"*The number of tide gauges taken into consideration for AMM15 and AMM7 is the same, therefore the coastal data, ….*"
The tide gauges data are from BODC
(https://www.bodc.ac.uk/data/hosted_data_systems/sea_level/uk_tide_gauge_network/) and from the North West Shelf Operational Oceanographic Service, NOOS, data portal
(http://noos.eurogoos.eu/). All the data are from tide gauges.
We have included these two web sites into the manuscript.

**P15.13 Suggestion to ease readability: in one of the maps, for instance in Figure 1, the authors should detail all the geographical names mentioned in the text (i.e. German Bight). This reference to geographical features will ease the reading of the paper to those potential readers not familiar with the regional geography.**

Thank for the comment, done

[Figure]

**P16.5 The authors shown in the paper (section 4.1.2) some results of the model validation with a HF Radar system. The results are only for 1 month (March 2017). If the authors have 2 years model runs, why do they perform/show a so short temporal coverage of the model- HF Radar validation? Due to observation availability? Please, explain reasons in the manuscript.**

We decided to focus our high resolution model validation on small areas and short time period, for both the glider and the HR radar observations. Both observations could be available for longer period and in different areas, but we have decided to stay focus on a short period to understand the impact of the high resolution and what is an adequate protocol to assess the quality of a high resolution model. As written in the manuscript at P1211 " …*have the benefit of providing an understanding of the impact of the high resolution locally on small area and short time scales*". This validation will be extended to assess the future evolution of the AMM15 system, since we proved it's useful and complementary to the standard validation protocol.

**Figure 6 shows the results of the model-HFRadar validation. In this figure, it is shown some statistics fields (RMSD, Bias, Veering) limited to the HF Radar spatial coverage. However, the reader have no information about the number of observational data that support these statistics. Do the HF-Radar system provide exactly the same number of observations everywhere? If yes, please detail what gap filling methodology is being applied. If not, please, show the % of HF-Radar data availability. I guess the 3 names referred in this figure 6 corresponds to the HF Radar sites. Please, detail in the Figure Caption.**

Thanks for this comment, we added the requested information in the manuscript,

"*One month, March 2017, of HF radar surface current velocity data were used to compare AMM7 and AMM15 in the German Bight where the bathymetry is shallow (Figure 4) and AMM15 is expected to performed better. The total surface velocity data from the COSYNA (Coastal Observing System for Northern and Arctic Seas) observing network (Gurgel et al., 2011), available through the EMODnet Physics data portal, are computed from radials of three HF radars installed on the islands of Sylt and*

*Wangerooge, and in Büsum (as shown on Figure 6). Data are averaged every 20 minutes on a grid of resolution of ~3 km. At the operating frequencies used, the total surface velocities represent an integrated velocity over a depth between 1 and 2 m. Relative error provided with the dataset was used to keep only data with error smaller than 15%. Model output were interpolated at the time and locations when and where observations were available to avoid applying gap-filling technics. Temporal coverage over the domain is larger than 75% everywhere except along the base line between Büsum and Wangerooge where the temporal coverage is ~29%."*

Yes, the three names correspond to the HF radar sites, thanks for this comment. The caption of the picture includes now this information.

**P18.13 The in-situ measurements are from buoys and ships of opportunity. Please, detail if "buoys" means fixed moorings, surface drifters or ARGO profilers.**

Thanks for this comment, we have corrected the manuscript as follow:
**"***The in-situ measurements are from different instruments, as detailed in* **Error! Reference source not found.***.*"

**P19.4 A Butterworth filter. Please, explain in more detail or add a reference.**

Thanks for the comment. We have modified the manuscript as follow, including a reference and enhancing the explanation:

*"A Butterworth filter (Butterworth, 1930) has been applied to the hourly model and observed SST data, using a cut-off for the filter at 5 days which removes the large scale synoptic and seasonal signals, leaving the internal dynamics and the wind driven signals, as well as the tidal frequencies".*

S. Butterworth, "On the Theory of Filter Amplifiers," Experimental Wireless and the Wireless Engineer, Vol. 7, 1930, pp. 536-541.

**P19.20 AMM7 and AMM15 models provide very similar values of SST, probably due to the data assimilation of SST that brings models close to the observations. Can the authors include in the paper any SST timeseries analysis as the one here shown for the 3 proposed sites, but in a station, whose SST observational data would not be assimilated? See some independent validation would certainly be of interest for readers and potential users of the model products.**

Thanks for the comment. Since SST has a very good satellite data coverage, it's not easy to exclude one single mooring from the set of observation assimilated and consider that observation completely independent. This is a typical dilemma while running an operational system. We are trying to assimilate all the available data to improve the quality of our products, but this implies that we reduce significantly the number of independent observations available for the validation. Due to our choice, we don't have a time series for a non -assimilated mooring.

**Figure 9. It is quite remarkable the overestimation of the 12-h energy peak in AMM15. Any relation with the harmonic bias in M2? It is also interesting the notorious AAM15 peak around 6-h. Can the authors comment on it? Any explanation? May it be linked to the meteorological forcing? (different in both model systems).**

The power spectra shown in Figure 9 is for the FINO 3 buoy (number 2 in Figure 4). The buoy is in the German bight, where the bathymetry is shallow (~20m). The 12h energy peak overestimation is remarkable in SON (wrongly marked as DJF in the manuscript, now corrected), at the end of the summer when probably the two models have different stratifications. The water column is moved by the tides (M2 in the predominant tide) and this could bring to differences in the SST variability. The stratification is this area could also be enhanced by the fresh water contribution of two major rivers, Elbe and Weser. This hypothesis is supported also from the analysis of the map of SST gradients (not shown in the paper) where AMM15 shows stronger gradients than AMM7. Further studies are needed to understand better the SST variability in AMM5.

**P23.1 E-Hype. What is E-Hype? No mention to this name in the section where forcing are described. Please, introduce complete name of the source or reference.**

Thanks for the comment. E-Hype is the hydrological model for the European areas developed by the Swedish Meteorological and Hydrological Institute (SMHI)**,** http://hypeweb.smhi.se/explore-water/geographical-domains/#europehype) We have included this information in the manuscript at P23.1 but not in the section where forcing are described because neither AMM15 nor AMM7(in the trial run described in this study)  are forced by E-Hype in the experiments described in this study. AMm7 was forced by E-Hype in a version in operation before April 2017.

**P24.24 Please, check the date: 23rd March or 23rd May (as referred in the Figure 12 caption; in this Fig 13 caption, correct typo: 23rt).**

Thanks, done.

**In Section 4.4.4 it is discussed about model differences in MLD and it is referred to the Figure 15, where only the MLD computed from the glider is depicted.  Why the MLD computed from the models are not shown in the panels Glider-MM15 & Glider-MM7 together with the one derived from the glider data? Include the MLDs from both models in the plot can enhance the analysis in this section dedicated to MLD.**

Thanks for the comment, we have added the model MLD to these figures (Yellow line for AMM15 and AMM7 respectively. The black line represents the MLD from the observations).
We added this information in the manuscript, P27.17:
*"...with AMM15 and AMM7 in the corresponding locations (yellow line in Figure 15)*" and in the caption of the figures.

[Figure]

**Section 4.5 is devoted to show some results from currents compared with HF-Radar data. As in the previous case for the tides, only a month of data (March 2017) is shown. Please, justify why a so short temporal coverage for the validation.**

Please see answer to comment P16.5

**Figure 16 shows monthly values of the HF-Radar and from the 2 models, interpolated to the observational field. However, no information on how many observations support the resulting monthly value is provided. Please, include the % of data availability for the month shown. It will be also useful to have some information on the validation of the HF-Radar measurement, as well as on the gap filling methodology used (if someone is used).**
**The explanation/discussion of the comparative results is quite poor. Please, provide some more description of the features depicted. For instance, it will be interesting that the authors describe the high currents feature existing in front of Wang and Busum stations, reproduced by the AMM15, but not for the AMM7 model. Likewise, any explanation or comment about possible border effects in the HF-Radar field shown would also be pertinent. Can the authors ensure that all the high currents depicted at the border of the HF Radar coverage are reliable? Please, include some info in the text (a reference would also help) on the existing validation of the HF Radar data used and about the possibility of border effects in the observational data used.**

Thanks to this comment we realised that we used in the manuscript the map of velocities before the cleaning of data instead of after. The reviewer is right, we can't ensure that the high currents depicted at the border of the HF radar coverage are reliable. We substituted Figure 16 with the corrected Figure.

[Figure]

We have also added the following information in the manuscript:

"*The HF radar surface currents were also used to investigate the sub-tidal circulation in the German Bight. The strong tidal signal in the shallow German Bight results in Kelvin waves propagating eastward on the southern boundary along Germany and northward at the eastern boundary along Denmark. However, this cyclonic circulation may not dominate as other processes are also influencing the circulation such as topographic effects from the shallow basin, wind and stratification resulting from freshwater input mostly from the Elbe and Weser river discharge. Wind tends to also produce a residual cyclonic circulation (Schrum 1997, Dick et al 2001, Port et al 2011). During the month of March 2017, a weak cyclonic circulation was observed in the mean HF radar surface currents along the German and Danish coasts (Figure 6). It is also observed in the AMM15 simulations and as a weaker flow in AMM7.  The strong flow out of the Elbe estuary is evident in AMM15 currents pattern, even if shifted to the west. AMM7 shows an intensification of its currents in this area, but with a speed much smaller than the observations and AMM15 (Figure 6). Generally….*"

Yes, we added the following references:

Dick S, Eckard K, Müller-Navarra S, Klein H, Komo H: The operational circulation model of BSH (BSHcmod)— model description and validation. Berichte des Bundesamtes für Seeschifffahrt und Hydrographie (BSH) 29, BSH, 2001.

Port, A., Gurgel, K. W., Staneva, J., Schulz-Stellenfleth, J., & Stanev, E. V.: Tidal and wind-driven surface currents in the German Bight: HFR observations versus model simulations. Ocean Dynamics, 61(10), 1567-1585, 2011.

Schrum C: Thermohaline stratification and instabilities at tidal mixing fronts. Results of an eddy resolving model for the German Bight. Cont Shelf Res 17(6):689–716, 1997.

**The analysis of the AMM7 & AMM15 model currents provided from P 28.14 till the end of the Section 4.5 (including reference to Figure 17) is not referred to any model validation. It is not used any observational data source used as reference. Therefore, I would suggest taking this analysis out from this Validation section.  I found the analysis interesting, and it illustrates quite well the dynamical differences existing between both model solutions.  If the authors want to keep this analysis in the manuscript, I would suggest moving this part of the text and the figure to the end of the Section 3 (where Trial experiments are described). This analysis of the dynamical patterns obtained gives a good idea of how different the 2 model solutions are and it may give a good introduction to the reader to the validation results that come later in Section 4.**

Done, as described at the very beginning of this document.

5.Conclusions

**P30.7 typo: temporal**

Corrected

**Please, include in the conclusion section some reference to the Data Assimilation performed in the AMM15 system, with mention to potential future plans to enhance the assimilation process (and very specially for SLA on the shelf).**

Thanks for this comment. We added in the manuscript the following paragraph:

"*The assimilation scheme used in AMM15 is broadly unchanged from that used in AMM7. While the short correlation length-scale is now ~5km (compared to ~20km), the observation and background error covariances, and the observation types assimilated, remain unchanged. In this initial implementation of AMM15 we have not attempted to improve the use of observations in the assimilation scheme. We are currently investigating how to adapt our assimilation scheme to assimilate SLA observations in stratified water and will be re-estimating the observation and background error covariances for this new higher resolution system.*"

**P31.18 The AMM15 ocean (system?).**

Corrected

---

## Author Comment (AC2) · 2 Jul 2019

Manuscript title: *"The impact of a new high-resolution ocean model on the Met Office North-West European Shelf forecasting system". by M. Tonani et al.*

**Bold: referee's comment**
Not bold: author's answer

The referee's comments are copied in this document for ease of reading.

General comment:

**The paper "The impact of a new high-resolution ocean model on the Met Office North-West European Shelf forecasting system" presents in a really useful and interesting way the main components of the high resolution regional ocean forecasting system and the validation protocol and results. Main novelties and innovative works in this study concern the high resolution of this regional forecasting system including data assimilation of the main available observations. As mentioned by the authors, it seems difficult to exhibit really significant improvements link to the higher resolution especially because the validation protocol is based on standard comparison between model and observations even if authors used specific high resolution observations based on glidersor HF radars. Nevertheless the study present an exhaustive comparison to available observations (assimilated or not) and validation diagnostics for most of the physical variables, these information are really useful for users of these operational forecast products and for developers of ocean forecasting system. I recommend the publication of this paper if the following minor revisions are taking into account in the final version.**

1.Introduction

1. **It could be useful to have a schematic view of the operational schedule of the system. The figure 2 with more information for example**

Thanks for this suggestion. We had added this information, the new figure 2 is:

[Figure]

2.      **Could you provide more precise information on the number of observations assimilated in the system thanks to the chosen assimilation cycles?**

From the manuscript:

*"The system runs forecast cycle every day to provide 6-day forecast …. By assimilating observations in this way, the FOAM system incorporates information from considerably more observations than would be available in near-real time with a single 24-hr window, due to the addition of late-arriving observations".*

The timeliness of the NRT observations could vary and be delivered with more than 24hr delay. For SST the delivery is usually within the 24-hr, therefore the impact is effectively zero for SST. NRT analysis (0h-24h) and Best estimate (24h-48h) have almost the same number of observations. The number of sub-surface profiles of temperature and salinity instead increases by ~15% by taking two days assimilation window instead of one. Given the low number of profiles this could be significant.

The number and quality of SLA observations increase in the file used for the Best estimate compared to the one available for NRT analysis. This is due to the production process of the SLA data, As described in Figure 2 in the Product User Manual (http://resources.marine.copernicus.eu/documents/PUM/CMEMS-SL-PUM-008-032-062.pdf) of this product.

The value available for the NRT analysis cycle are marked in orange in Figure 2 and are produced using altimeter fast-delivery input (Operational Geophysical Data Record, OGDR, or L2P Near Real Time). The value available for the Best Estimate cycle instead, in yellow, are produced using the altimeter real time data (Interim Geophysical Data Record, IGDR, or L2P Short Time Critical). The fast delivery input data have less measurements and lower accuracy.

Figure from CMEMS PUM:

[Figure]

**Figure 2: Data delivery flow for Global NRT SL-TAC products**

Providing an estimation of the different number of observations and quality it's complicated by our assumption to assimilate data only where the depth is higher than 700m. An estimation of the differences between the SLA data available for the Best Estimate and the NRT Analysis are:

[Figure]

*Figure: SLA observations available for the Best Estimate (left panel) and for the NRT Analysis (right panel). The value along the track represents the time associated with the measurements. The time is represented in Julian day. All the values taken after ~21:00 are not included in the NRT but are in the Best Estimate Analysis. Also measurements taken before 21:00, could be missing in the NRT data (e.g.: green line between 57 N- 61N).*

3.      **You mentioned the on going development of physic-biogeochemistry coupled system and the operational constrain. It's not the topic of the paper, but I suggest there is too much or not enough information for readers. Could you add few words about the time constrain and what kind of development is expected to reach the goal.**

The first version of the biogeochemical model coupled at 1.5km was made available at the end of year 2018. The preliminary tests required an extensive use of computational resources, not compatible with the operational requirements. One day (24 hours) of coupled model run required ~ 2.5 hours. The production of a full forecast cycle would have been around ~ 25 hours, for the 2 days with data assimilation and the 6 -day forecast. This number are prohibitive for a daily production cycle. These tests are running on the Met Office HPC – Cray XC40 super computer using 48 nodes and 1536 processors.

R&D activities are trying to improve the use of the resources and investigate different solutions for the coupling like a coarser time step or grid for the biogeochemical model.

The manuscript has modified:

"*The upgrade of the NWS system to AMM15 does not yet include the biogeochemical component as the computational cost is prohibitive, because the production time exceeds the 24-hr for a full hindcast-forecast cycle.*"

2.System Development

2.1.Core model Description

1. **One specificity of the model configuration is the vertical coordinate system based on z\*-σ. There is no justification in the description paragraph concerning the number of vertical levels which is the same than in the lower resolution system. Is there theoretical or experimental justification to reduce the rmax coefficient to 0.1 in this high resolution configuration and what is the expected impact (except the numerical stability)?**

The major aim of this model configuration is to resolve the Rossby Radius on the shelf, therefore the focus was on increasing the resolution from 7 to 1.5 km. more than increasing the vertical resolution.

The number of vertical levels is the same because the focus of this model is on the shelf (depth < 200m), where 51 z-sigma levels are enough for proving a very high vertical resolution. The resolution is of the order of 20cm the shallower part of the model domain, where the minimum depth is 10. More levels will increase the model vertical resolution in the deepest part of the domain, not on shelf (Siddorn et al., 20016). Another possible approach is using vertically adaptive vertical coordinates so that you focus resolution on the thermocline. This is done in other models but not here and will be considered in the future configurations.

All the technical details of the implementation and the validation of the model, without data assimilation, are presented in Graham et al., 2018 that is the precursor work to this paper.

The justification for the rmax choice is Graham et al. 2018a:

"With terrain-following coordinates, large slopes between adjacent grid cells can lead to pressure gradient errors. To reduce such errors, vertical cells can be masked over slopes which exceed a specified value, rmax, where $r = (h_i - h_{i+1})/(h_i + h_{i+1})$, and $h_{i,i+1}$ are adjacent bathymetry points. Terrain-following coordinates are fitted to a smoothed envelope bathymetry, with the level of smoothing based on the chosen rmax value. In regions where the smoothed model levels become deeper than the input bathymetry, these levels are then masked. Thermax value was chosen here to be 0.1. This is a lower value than used in previous configurations. However, with increased resolution, the model bathymetry is rougher, resolving steeper gradients and canyons along the shelf break. This value was then chosen to ensure stability in the configuration without the need to smooth the input bathymetry".

2. **You impose a minimum of 10m depth on the bathymetry (this characteristic is also mentioned in the conclusion as a limitation), could you justify this choice, is only due to model stability?**

This is down to the tidal limits and lack of wetting and drying. 10m ensures that no locations dry out (e.g. Bristol channel).

This information is now added in the manuscript:

"*The model minimum depth is forced to be 10m, due to the tidal limit and lack of wetting and dry. This choice ensures that no locations dry out, due to the tides.*"

3. **How do you justify such difference (2 orders of magnitude) between the diffusion coefficient on tracer and advection?**

These values we chosen over a series of sensitivity tests. We aimed to keep diffusion parameters as low as possible (due to resolving processes at higher resolution), opting for bi-laplacian diffusion along model levels primarily to ensure stability. For momentum, the value was chosen to account for processes that are still missing (e.g. smaller scale frictional processes). For tracers, we initially started with the same order of magnitude. However, these results appeared to be too diffusive, so following tests opted for a less diffusive value, but one that would still provide stable conditions under long simulations.

**2.1.1 Boundary and surface forcing**

1. **Could you add in the table 2 information concerning the difference of solar flux penetration in the two configurations and information on the tidal forcing at lateral boundaries**

Yes, thanks for the correction. The tidal forcing information have been added to table 2. The differences concerning the solar flux penetration are in Table 1.

Updated Table 2:

| Forcing | AMM7 | AMM15 |
|---|---|---|
| **Surface forcing** | Met Office Global Unified Model (MetUM) Atmospheric model NWP analysis and forecast fields, calculated in the MetUM using COARE4 bulk formulae (Fairall et al. 2003). | ECMWF Integrated Forecasting System (IFS)-Atmospheric Model High Resolution (HRES) operational NWP forecast fields using CORE bulk formulae (Large and Yeager 2009) |
| **Surface forcing resolution** | Horizontal grid: ~10 km (2560 x 1920 grid points) Frequency: 3 hourly mean fluxes of long and short wave radiation, moisture, 3 hourly mean air surface temperature but hourly 10m winds and surface pressure | Horizontal grid: ~14 km (0.125°x0.125°). Frequency: 3 hourly instantaneous 2m dew point temperature, surface pressure, mean sea level pressure, and 2m air temperature. 3 hourly accumulated surface thermal and solar radiation, total precipitation, and total snow fall. |
| **River run-off** | Daily climatology of gauge data averaged for 1950–2005. Climatology of daily discharge data for 279 rivers from the Global River Discharge Data Base (Vörösmarty et al., 2000) and from data prepared by the Centre for Ecology and Hydrology as used by (Young and Holt, 2007). | Daily climatology of gauge data averaged for 1980–2014. UK data were processed from raw data provided by the Environment Agency, the Scottish Environment Protection Agency, the Rivers Agency (Norther Ireland), and the National River Flow Archive (personal communication by Sonja M. van Leeuwen, CEFAS, 2016). For major rivers that were missing from this data set (e.g. along the French and Norwegian coast), data have been provided by the same climatology used by AMM7 (Vörösmarty et al., 2000 and Young and Holt, 2007). |
| **Tidal constituents** | M2, S2, N2, K2, K1, O1, P1, Q1, M4, MS4, L2, T2, S1, 2N2, MU2, NU2 (15) from a tidal model of the North-East Atlantic (Flather, 1981). | M2, S2, N2, K2, K1, O1, P1, Q1, M4, MS4, MN4 (11) from Topex Poseidon cross-over solution (Egbert and Erofeeva, 2002; TPX07.2, Atlantic Ocean 2011-ATLAS). |
| **Lateral boundaries** | Met Office FOAM North Atlantic (1/12°; 6 hourly fields) and CMEMS Baltic Sea (2km, 1 hourly fields). | |
| | AMM7 and AMM15 have Atlantic and Baltic boundaries in a different geographical location. | |

**2.2 Assimilation method**

**Some information are missing in the description:**

**1. How is implemented the IAU method?**

We have rephrased the manuscript sentence: "*The increments are applied to the model fields at each time-step using the incremental analysis update procedure (IAY, Bloom et al. 1996)*" with the following: "*After the assimilation step, the model is re-run for the same period with a fraction of the increments applied to the model fields at each time step (the incremental analysis update procedure, Bloom et al. 1996)*".

We hope this clarifies to the readers how the IAU method is implemented.

**2. What is the SLA bias correction?**

We have expanded page 7, line 14 adding the following text: " *The Met Office implementation of NEMOVAR includes bias correction scheme for both SST and altimeter data. The SST bias correction aims to correct for biases in the observed SST due to the synoptic scale atmospheric errors in the satellite retrievals, while for SLA we apply a slowly-evolving bias correction to correct for errors in the MDT (Lea et al. 2008)".*

3. **How do you use the 2 correlation length scale in the assimilation scheme? Do you perform 2 analysis?**

There is only one analysis. The correlation operator used in the specification of the background errors within NEMOVAR is a linear combination of functions with different length-scales (see Mirouze et al. 2016). This allow us to define a correlation operator that features high correlations within a short scale and weak correlations at large scales.

We added this reference is the manuscript.

Mirouze, I, Blockley, E. W., Lea,D.J., Martin, M.J., Bell, M.J.: A multiple length scale correlation operator for ocean data assimilation, Tellus A 2016, 68, 29744, http://dx.doi.org/10.3402/tellusa.v68.29744, 2016

4. **In table 4, what are the differences between the 2 in situ data sources. How do you manage observation available in the two data bases?**

The differences are in the data format, distribution protocol, timeliness. Some of the data sources are in common and therefore we perform a duplicate check before ingesting the observations in the analysis.

5. **In table 4, there is no information on the mean dynamic topography used to assimilate the SLA.**

We use the CNES-CL09 mean dynamic topography (MDT, Rio et al. 2011) to calculate observations of the SSH from the observed SLA which can be compared to our model SSH fields. We have added this information to table 4.

| Data Assimilation | AMM7 | AMM15 |
|---|---|---|
| **NEMOVAR version** | V3 | V4 |
| **SST bias correction scheme:** | Offline observations-of-bias scheme. Reference dataset: in-situ. | Variational scheme in addition to observations-of-bias. Reference datasets: in-situ (drifters only) and VIIRS satellite data. |
| **Correlation operator short scale: 3-times grid scale** | ~20 km | ~5 km |
| **Mean Dynamic Topography** | CNES-CL09 mean dynamic topography (MDT, Rio et al. 2011) | |

6. **There is no information on methodology applied to assimilate the SLA in the model including tides.**

We added the following to page 7, line 12: *"… as detailed in Table 4. The SLA observations assimilated in this model are provided through CMEMS and include the corrections necessary to add back the signals due to tides and wind and pressure effects necessary for use with a wind and pressure forced, tidal coastal model (King et al. 2018)"*.

**2.3 Operational production**

1. **How is computed the QC error threshold for the observations?**

The QC error threshold for the observations is defined on the base of the model-observation difference and varies with depth. Temperature and salinity have a different threshold error. The details on the background check are described in Ingleby et al., 2007. We added this reference to the manuscript. We corrected also the typo 1/3 with "*1/2*".

Ingleby, B., Huddlestone, M.: Quality control of ocean temperature and salinity profiles — Historical and real-time data, Journal of Marine System, Vol. 65, Issue 1-4, pp. 158-175, https://doi.org/10.1016/j.jmarsys.2005.11.019, 2007.

2. **You provide output fields on a standard vertical grid, how do you provide the information at the surface (0m)? Is there a specific extrapolation to the surface?**

The surface level is the model first level, we don't apply any specific extrapolation at the surface. We have substitute 0 with "*surface*" in the manuscript to avoid confusion.

3. **Additional information concerning computational resources for this operational system could be useful (number of CPU, computer characteristics…)**

These operational systems are running on the Met Office HPC – Cray XC40 super computer. The information in terms of number of nodes and processors used by each component of the system are in the following table:

| System | Component | # of nodes | # of processors |
|--------|-----------|------------|-----------------|
| AMM7 | NEMO | 8 | 256 |
| | XIOS | -- | -- |
| | NEMOVAR | 2 | 64 |
| AMM15 | NEMO | 48 | 1536 |
| | XIOS | 8 | 256 |
| | NEMOVAR | 48 | 1536 |

XIOS is for the I/O of NEMO. The small size of AMM7 model grid doesn't require dedicated nodes for this task.

We have added this information in the manuscript.

4. **More information could be added on figure 2 as for example, the observations, the atmospheric forcing, the restart and the assimilation and forecast sequence.**

We have increased the number of information in figure 2, providing more details on the forecast production cycle. (see answer Question 1, Introduction).

**4.Validation**

**4.1Tides**

1. **M2 is the dominant tidal signal and probably the most important in an operational system for applications, user needs …. One unexpected result increasing the resolution is perhaps the degradation of the mean M2 solution. It will be important in this section to discuss this point and highlight origin of this degradation.**

AMM15has a higher mean error (few cm higher than AMM7) but a better RMSD than AMM7. This is explained in Graham at al. 2018:

"For AMM7, while the RMSE has a similar magnitude to AMM15, compensating errors in both amplitude and phase are found around the UK, reducing the apparent mean bias."

Yes, the referee is correct, it is important to improve the tidal forcing of AMM15, in terms of tidal constituents and atlases. Research activities are ongoing to validate the impact of using a different model, FES2014, with many more tidal constituents.

**4.2Sea Surface Height**

**The section concerning SSH, as it is, is not really useful and could be removed. But as the SSH is assimilated in the system it's important to quantify impact of these observations. I suggest to add few diagnostics in comparison to SSH as for example:**

**Statistic/comparison with altimetry in open ocean where observation are assimilated. Along track comparison could be performed. It's important to understand in the paper why SLA is assimilated in the system**

**Spatial power spectra to quantify spatial resolution of the system**

**Variability or eddy kinetic energy**

The point of this short section is not to quantify the impact of assimilating SLA, this was done in King et al. 2018, but to verify that we can achieve similar accuracy (in terms of bias and RMSD) with the higher resolution model. The current altimeter assimilation is limited and there are plans to extend the assimilation into the shallow water regions which are tidally dominated.

We describe in the paper the procedure for the validation of the trial experiments for the pre-operational implementation of this system. The evolution of the model and data assimilation components are those described in Graham et al 2018 for the model and King et al. 2017 for the data assimilation.

**4.3 Sea Surface Temperature**

**Temporal variability from seasonal cycle to high frequency is validated comparing model output to satellite observations and in situ time series. As expected there are few differences between the two models, main difference between the models being the horizontal resolution, even if the authors exhibit interesting higher frequency processes in the high resolution system. Even if it is not feasible with the observations why any spatial power spectra (or other diagnostics) has been performed to quantify differences between the 2 models?**

As the referee is pointing out, it is difficult to identify an SST L4 product with a resolution comparable or higher than AMM15. We have done seasonal gradient maps from AMM7, AMM15 and OSTIA (not shown in the paper) and it's difficult to validate the increased variability of AMM15. The power spectrum plots shown in the manuscript show bigger differences between AMM15 and AMM7 during the autumn, probably due to the different stratification of the two models in that area. We copied here the details preferee#2 on figure9:

The power spectra shown in Figure 9 is for the FINO 3 buoy (number 2 in Figure 4). The buoy is in the German bight, where the bathymetry is shallow (~20m). The 12h energy peak overestimation is remarkable in SON (wrongly marked as DJF in the manuscript, now corrected), at the end of the summer when probably the two models have different stratifications. The water column is moved by the tides (M2 in the predominant tide) and this could bring to differences in the SST variability. The stratification is this area could also be enhanced by the fresh water contribution of two major rivers, Elbe and Weser. This hypothesis is supported also from the analysis of the map of SST gradients (not shown in the paper) where AMM15 shows stronger gradients than AMM7. Further studies are needed to understand better the SST variability in AMM5.

**4.4 Water Column**

**On figure 10 larger bias and larger differences between AMM15 and AMM7 is located at 100m depth. Is it linked to Mediterranean water? How do you explain this difference if the two configurations have the same constrains at the boundary and assimilates the same observations?**

The large AMM7 bias is due to the vertical level discretization. With terrain-following coordinates, large slopes between adjacent grid cells can lead to pressure gradient errors. To reduce such errors, vertical cells can be masked over slopes which exceed a specified value, rmax. (Graham et al. 2018). AMM15 has a smaller value of Rmax ( 0.1) than AMM7 (0.3). The vertical discretization of AMM7, when the slope is too large over the shelf break, could end up with cells connected horizontally that are very different in vertical position. This means that the model is mixing in the horizontal sense water from two very different depths. Reducing the allowed slope, as it is in AMM15, prevents this artificial (or reduces) diapycnal mixing.

This is why AMM15 has a reduced bias at depth compared to AMM7.

**4.4.2 Moorings German Bight**

**Few more information or hypothesis will be useful to explain some description. –"The high frequency is better reproduced". Do you compute the correlation between the time series ? It's not clear on figure 11.**

We added this sentence:

 "The improvement is more evident in the summer (JJA) when AMM7 has a fresh bias of ~0.5 PSU while AMM15 has values very close to the observation."

No, we didn't compute the correlation between the time series.

**-"at the bottom AMM15 is more accurate". Why? Is it link to the bathymetry or link to vertical projection of increments?**

This is probably due to both. AMM15 bathymetry is more accurate and the higher resolution improves the representation of the model bottom, especially in these shallow areas.

**-Table 8 : what is the depth of the bottom of each Buoy position?**

Thanks for this comment, we added the depth of each buoy in table 8. The depth of these moorings varies from 18 to 35 m.

The updated table 8:

| Buoy [bottom depth] | Temperature (C°) | | | | | | | |
|---|---|---|---|---|---|---|---|---|
| | Surface | | | | Bottom | | | |
| | RMS Difference | | Mean Errors | | RMS Difference | | Mean Error | |
| | AMM7 | AMM15 | AMM7 | AMM15 | AMM7 | AMM15 | AMM7 | AMM15 |
| 1 Fino1 [25m] | 0.32 | **0.21** | **0.03** | -0.05 | 0.31 | **0.21** | 0.07 | **-0.03** |
| 2 Fino3 [18m] | 0.38 | **0.37** | **-0.02** | -0.04 | 0.96 | **0.59** | -0.38 | **-0.24** |
| 3 NsbII [35m] | 0.30 | **0.25** | 0.12 | 0.12 | 0.59 | **0.49** | **-0.13** | -0.14 |
| 4TWEms [30m] | 0.28 | **0.26** | 0.13 | **-0.02** | 0.28 | **0.16** | 0.11 | **0.00** |
| 5 UFSDeBucht [20m] | 0.50 | 0.50 | 0.10 | **0.01** | 0.95 | **0.75** | -0.31 | -0.33 |
| Mean value | 0.36 | **0.32** | 0.07 | **0** | 0.62 | **0.44** | **-0.13** | -0.15 |

| Buoy [bottom depth] | Salinity (PSU) | | | | | | | |
|---|---|---|---|---|---|---|---|---|
| | Surface | | | | Bottom | | | |
| | RMS Difference | | Mean error | | RMS Difference | | Mean Error | |
| | AMM7 | AMM15 | AMM7 | AMM15 | AMM7 | AMM15 | AMM7 | AMM15 |
| 1 Fino1 [25m] | 1.17 | **1.02** | 0.97 | 0.97 | 1.10 | **1.02** | 0.95 | **0.95** |
| 2 Fino3 [18m] | 1.06 | **0.73** | **0.35** | 0.48 | 0.90 | **0.62** | 0.53 | **0.38** |
| 3 NsbII [35m] | 0.33 | **0.22** | 0.20 | **0.03** | 0.37 | **0.17** | 0.26 | **0.03** |
| 4 TWEms [30m] | 1.05 | **0.51** | 0.85 | **0.29** | 1.08 | **0.45** | 0.89 | **0.26** |
| 5 UFSDeBucht [20m] | **0.99** | 1.07 | **0.55** | 0.87 | 1.08 | **1.02** | 0.86 | 0.90 |
| Mean value | 0.92 | **0.71** | 0.58 | **0.53** | 0.91 | **0.66** | 0.70 | **0.51** |

**-Figure9: why there is no model information in October? Add the correlation on the figure**

Thanks for this comment, we have done a new picture, covering only the period January-October to avoid confusion. There are no measurements from the NsbII mooring in October, due to maintenance or malfunction of the sensor, the comparison model-observation is not possible. We double checked the other moorings and none of them is without interruptions.

We have added this information in the label of the new picture.

[Figure]

"*Figure 1: Sea surface salinity at the NsbII mooring for January-September 2017. Observations for October-December are not available*"

**4.4.3 Glider transects**

**Could you precise if the glider observations are assimilated or not in the system?**

No, the glider observations are not assimilated in AMM7 nor AMM15. Both systems assimilate glider observations but not the profiles from the MASSMO4 2017 campaign.

**4.4.4 Mixed Layer depth**

**I suggest adding the mixed layer depth for AMM15 and AMM7 on figure 15 for example.**

Thanks for the comment, we have added the model MLD to these figures (Yellow line for AMM15 and AMM7 respectively. The black line represents the MLD from the observations).

We added this information in the manuscript:

"*...with AMM15 and AMM7 in the corresponding locations (yellow line in Figure 15)*" and in the caption of the figures.

[Figure]

**4.5Currents**

**The comparison with HF radar observations is very useful and seems to be more relevant to compare high and low resolution model outputs. I suggest adding the statistics (mean, rms, correlation on amplitude and direction) which seems to be encouraging for the high resolution model as it is explain in the text but without the figures.**

Thanks to this comment we realised that we used in the manuscript the map of velocities before the cleaning of data instead of after. We substituted Figure 16 with the corrected Figure

[Figure]

We added the following text to the manuscript:

*"One month, March 2017, of HF radar surface current velocity data were used to compare AMM7 and AMM15 in the German Bight where the bathymetry is shallow (Figure 4) and AMM15 is expected to performed better. The total surface velocity data from the COSYNA (Coastal Observing System for Northern and Arctic Seas) observing network (Gurgel et al., 2011), available through the EMODnet Physics data portal, are computed from radials of three HF radars installed on the islands of Sylt and Wangerooge, and in Büsum (as shown on Figure 6). Data are averaged every 20 minutes on a grid of resolution of ~3 km. At the operating frequencies used, the total surface velocities represent an integrated velocity over a depth between 1 and 2 m. Relative error provided with the dataset was used to keep only data with error smaller than 15%. Model output were interpolated at the time and locations when and where observations were available to avoid applying gap-filling technics.*

*Temporal coverage over the domain is larger than 75% everywhere except along the base line between Büsum and Wangerooge where the temporal coverage is ~29%."*

**Figure 17 is nice to exhibit differences between the 2 models. It could be even better to add map with high resolution observations on the same area. Is there any SLA, SST or ocean colour map that can be used to compare front and meso scale structure?**

We agree with the reviewer, but we are not aware of any satellite map at a comparable resolution of AMM15. CMEMS has several products but the resolution varies from 1/4 -1/8 of degree with the exception of the ocean colour data from OLCI or the Odyssee SST L4 product. The ocean colour data have several gaps and it could be very difficult to make a comparison. We tried to look at the SST L4 data from Odyssee but since currents like the Norwegian coastal currents and the Scottish coastal currents are mainly salinity driven there is no signal in the SST maps, at least not at the resolution of the currents of AMM15.

We followed the suggestion of the referee#2 and we removed this part from the validation. We moved it at the end of the trial description, adding a new section.

**5 Conclusion and future developments**

**Something is missing in the conclusion, even if it is not obvious to validate and quantify improvement link to the higher resolution a discussion on expected improvements and link with user needs on this domain will be useful**

We added the following sentence:

*"The users' benefit, using the newly improved European shelf product AMM15, will vary depending from their applications. Higher resolution currents fields with an improved representation of the coastal areas should improve the results of applications like drifting models simulating pollutant or oil spill dispersion and all the applications that need a high resolution currents field. All the acoustic applications, strongly depending on the density stratification and its variability, will benefit from these new products since they have a better representation of the water masses. A general positive impact is expected for most of the users like public bodies responsible for marine environmental regulation, aquaculture industries, marine renewable oil and gas industries."*

**Typo, figures or format correction**

1. **Section Boundary and Surface Forcing should be 2.2 and then 2.3 Assimilation method, 2.4 operational system**

Done

2. **Table 4 is cited before table 3**

Corrected

3. **Conclusion I 7 spatial/temporal**

Corrected

---

## Author Response (AR2)

Dear topic Editor,

 I checked for the typo and other small mistakes. I reversed the colour range of the M2 amplitude and phase because I did a mistake and displayed in the previous version model-observation instead of observation- model.

A special thanks to the anonymous reviewers, the manuscript has been improved thanks to their comments/suggestions.

Kind Regards

Marina